# A p38α-BLIMP1 signalling pathway is essential for plasma cell differentiation

Jianfeng Wu[1,2,4], Kang Yang[1,4], Shaowei Cai[1,4], Xiaohan Zhang[1,4], Lichen Hu[1], Fanjia Lin[1], Su-qin Wu[2], Changchun Xiao[1], Wen-Hsien Liu ●[1] ✉ & Jiahuai Han ●[1,2,3] ✉

Plasma cells (PC) are antibody-secreting cells and terminal effectors in humoral responses. PCs differentiate directly from activated B cells in response to T cell-independent (TI) antigens or from germinal center B (GCB) cells in T cell-dependent (TD) antigen-induced humoral responses, both of which pathways are essentially regulated by the transcription factor BLIMP1. The p38 mitogen-activated protein kinase isoforms have already been implicated in B cell development, but the precise role of p38α in B cell differentiation is still largely unknown. Here we show that PC differentiation and antibody responses are severely impaired in mice with B cell-specific deletion of *p38α*, while B cell development and the GCB cell response are spared. By utilizing a *Blimp1* reporter mouse model, we show that *p38α*-deficiency results in decreased BLIMP1 expression. p38α-driven BLIMP1 up-regulation is required for both TI and TD PCs differentiation. By combining CRISPR/Cas9 screening and other approaches, we identify TCF3, TCF4 and IRF4 as downstream effectors of p38α to control PC differentiation via *Blimp1* transcription. This study thus identifies an important signalling pathway underpinning PC differentiation upstream of BLIMP1, and points to a highly specialized and non-redundant role for p38α among p38 isoforms.

Humoral immune response, characterized by high-affinity antibody production, protects organisms from pathogens and plays an indispensable role in vaccination[1,2]. In response to T cell-dependent (TD) antigens, naïve B cells undergo extensive proliferation and form germinal center (GC), where they become germinal center B (GCB) cells, undergo class-switch recombination (CSR), somatic hypermutation (SHM), affinity maturation, and differentiate into plasma cells (PC) or memory B cells[3,4]. PCs are the terminal effector cells in humoral immune responses and their differentiation and function are tightly regulated by a network of transcription factors[5,6]. Among those transcription factors, BLIMP1 functions as the master regulator. Deletion of *Blimp1* completely blocked PC differentiation[7–9], and ectopic BLIMP1 expression induced immunoglobulin (Ig)-secreting PCs differentiation from B-cell lymphoma lines[10]. BLIMP1 expression is upregulated during PC differentiation, and BLIMP1 mainly functions as a transcription repressor to control PC differentiation and homeostasis by repressing the expression of genes critical for mature B cell identity and promoting the expression of genes critical for PC identity[11,12]. Several transcription factors, including NF-κB, STAT3, IRF4, AP-1 and E2A, directly bind the *Blimp1* locus and promote *Blimp1* transcription after B cell activation by BCR, pattern recognition receptors (PRRs) and signals from T cells[13–15]. Moreover, several kinases such as PI3K, Jaks, and ERK1/2 have been reported to regulate the activity of those transcription factors in this process[13,16,17].

[1]State Key Laboratory of Cellular Stress Biology, School of Life Sciences, Faculty of Medicine and Life Sciences, Xiamen University, Xiamen, Fujian 361102, China. [2]Laboratory animal research center, Xiamen University, Xiamen, Fujian 361102, China. [3]Research Unit of Cellular Stress of CAMS, Cancer Research Center of Xiamen University, Xiang'an Hospital of Xiamen University, School of Medicine, Xiamen University, Xiamen, Fujian 361102, China. [4]These authors contributed equally: Jianfeng Wu, Kang Yang, Shaowei Cai, Xiaohan Zhang. ✉e-mail: whliu@xmu.edu.cn; jhan@xmu.edu.cn

As the antibody-producing factory, each mature PC secrets thousands of antibody molecules per second and is characterized by expanded endoplasmic reticulum (ER)[18,19]. Importantly, ER expansion and induction of the unfolded protein response (UPR) are required for robust immunoglobulin protein synthesis during PC differentiation, as UPR upregulates chaperone expression and expands the ER network to facilitate proper folding and secretion of immunoglobulins[20,21].

It is well-known that the most frequent primary immunodeficiencies in human subjects are primary antibody deficiencies (PADs), and defect in plasma cell differentiation was one of the main causes of PADs or diffuse large B cell lymphoma (DLBLC)[22–24]. Conversely, dysregulated plasma cell generation often leads to production and accumulation of autoantibodies, which is an important cause of autoimmune diseases, such as systemic lupus erythematosus (SLE)[25,26]. Several drugs that inhibit plasma cell differentiation has been used for treating those autoimmune diseases in clinical[26–28].

p38 represents a highly conserved mitogen-activated protein kinase (MAPK) group and its prototype member p38α is one of the most extensively studied proteins. More than 20 clinical trials employ p38 group MAPKs as drug targets, including two clinical trials to treat COVID-19[29,30]. p38α participates in differentiation of various types of cells including immune cells. Using an in vitro system, B cell generation and lymphoid versus myeloid fate decision were found to depend on MEF2C-EBF1 transcription factors and p38 inhibitor can block this process[31]. Moreover, MEF2C was crucial for BCR-stimulated B cell proliferation and TD immune response and p38 inhibitor can inhibit BCR-stimulated MEF2C activity in these process[32,33]. In addition, p38α activation was required for CD40-induced gene expression in B cells[34]. However, B cell development in bone marrow and spleen was normal in *Mef2c*[fl/fl]*CD19*[cre] mice[32,33], and B cell development, activation and proliferation were normal in *p38α*[−/−] chimeric mice generated by ES-cell mediated RAG-deficient blastocyst complementation technology[35]. Thus, more systemic genetic studies are needed to define the roles of p38α and p38α-MEF2C axis in B cell development, GCB generation, PC differentiation and antibody responses.

In this study, we employ genetic approaches to show that p38α activation is required for the PC differentiation, but dispensable for B cell development, activation, proliferation, and GCB cell generation. We further show that TCF3, TCF4, and IRF4 function downstream of p38α to promote *Blimp1* transcription, and reveal that IRF4 is directly activated by p38α via phosphorylation. As dysregulated PC differentiation often leads to autoimmune disease, p38α could be a therapeutic target for treating those diseases.

## Results
### p38α is essential for plasma cell generation
We set out to investigate the physiological role of p38α in B cells by generating and analyzing mutant mice with B cell-specific deletion of *p38α*. *p38α*[fl/fl] mice were crossed with *CD19*[cre] mice, which express the Cre recombinase from the endogenous *CD19* locus[36]. *p38α*[fl/fl]*CD19*[cre] mice were viable and did not display any gross physical or behavioral abnormalities. Development of B cell subsets in the spleen and bone marrow of 5–6 weeks old mice showed almost no difference between *p38α*[fl/fl]*CD19*[cre] and *p38α*[fl/fl] mice (Supplementary Fig. 1a). Although the percentage of small pre B cells in bone marrow from *p38α*[fl/fl]*CD19*[cre] mice was less than that in WT mice, total cell number and frequency of peripheral B cells were not changed in *p38α*[fl/fl]*CD19*[cre] mice (Supplementary Fig. 1a). Similarly, no significant effect of *p38α* deletion, starting at early hematopoiesis in *p38α*[fl/fl]*Vav*[cre] mice, on the generation of B cell subsets was found (Supplementary Fig. 1b). These results suggest that *p38α* is dispensable in the generation of almost all B cell subsets in mice, and we did not further address the difference of small pre B cells.

To study the role of p38α in antibody responses, we immunized *p38α*[fl/fl]*CD19*[cre] mice and their WT littermates with OVA/Alum/LPS and analyzed GC reaction in the spleen 7.5 days later. The including LPS in the stimulation is to cause a strong immune response[37]. As shown in Fig. 1a, while the percentage and number of GCB cells were similar, *p38α*[fl/fl]*CD19*[cre] mice exhibited significant reduction in PCs. Consistent with the no change in the numbers of GCB cells in Fig. 1a, the structure and size of GCs in the spleen were comparable between immunized *p38α*[fl/fl]*CD19*[cre] and *p38α*[fl/fl] mice (Fig. 1b). Since the OVA/Alum/LPS immunization strategy often leads to extrafollicular PCs generation, we further examined antigen-specific GCB cells and plasmablasts (PB) in the spleen of NP-OVA/Alum-immunized *p38α*[fl/fl]*CD19*[cre] and their WT littermate mice. As shown in Fig. 1c, d, the percentage and number of NP-specific GCB cells were similar in *p38α*[fl/fl]*CD19*[cre] and *p38α*[fl/fl] mice, whereas NP-specific plasmablasts and NP-specific IgG1 antibody-secreting cells (ASC) were drastically reduced in NP-OVA/Alum-immunized *p38α*[fl/fl]*CD19*[cre] mice, in line with the results obtained from mice with OVA/Alum/LPS immunization (Fig. 1a). These results suggest that the regulation of PC generation by p38α most likely occurs at stages between GCB cells and PCs.

We then examined the role of p38α in antibody production and accumulation. In the absence of immunization, the basal serum level of IgG1 was lower than, whereas the concentrations of IgM, IgG2a, IgG2b, and IgG3 in *p38α*[fl/fl]*CD19*[cre] mice were similar to *p38α*[fl/fl] mice (Supplementary Fig. 1c). Correlated with these data, we also observed a decrease in the number of PCs in non-immunized *p38α*[fl/fl]*CD19*[cre] mice (Supplementary Fig. 1d). Those mice were then immunized with NP-OVA/Alum and serum concentrations of total NP-specific (anti-NP$_{29}$) IgM and IgG1, and high-affinity (anti-NP$_7$) IgG1 were determined by ELISA. *p38α*[fl/fl]*CD19*[cre] mice had significantly less production of total and high-affinity NP-specific IgG1 but the NP-specific IgM levels were similar between *p38α*[fl/fl]*CD19*[cre] and *p38α*[fl/fl] mice (Fig. 1e). Reduction in NP-specific IgG1 levels was also observed in secondary antibody responses of *p38α*[fl/fl]*CD19*[cre] mice (Fig. 1e). Meanwhile, NP-specific plasmablasts and NP-specific IgM, IgG2b, IgG3 ASCs were also significantly less in NP-LPS immunized *p38α*[fl/fl]*CD19*[cre] mice in comparison with *p38α*[fl/fl] mice (Supplementary Fig. 2a). On the other hand, immunization with NP-Ficoll, by which IgG3 is the dominant IgG form in antibody response, induced similar amounts of NP-specific IgG3 in *p38α*[fl/fl]*CD19*[cre] and *p38α*[fl/fl] mice (Supplementary Fig. 2b), suggesting that p38α in B cells was not involved in type II T cell-independent immune response. Consistently, PC generation was significantly decreased by *p38α* knockout in *p38α*[fl/fl]*CD19*[cre] mice after NP-LPS immunization (Supplementary Fig. 2a) but was comparable in NP-Ficoll-immunized mice (Supplementary Fig. 2c). Collectively, we could conclude that p38α in B cells is essential for the antibody responses to TD antigen or to type I T cell-independent antigen, and we then focused on the role of p38α in PC generation from GCB cells in this study.

### p38α promotes PC generation in a cell-intrinsic and kinase activity-dependent manner
To elucidate the function of p38α in PC generation at cellular and molecular levels, we performed PC differentiation in an in vitro culture system in which naïve B cells proliferate on top of a Balb/c 3T3 cell line stably expressing CD40L and BAFF (termed 40LB cells). In the presence of IL-4, B cells acquire a GCB phenotype (termed iGCB cells) after 4 days of culture. These iGCB cells can further differentiate into PCs (termed iPCs, that are mostly IgG1 and IgE producing cells) after 4 additional days of culture on 40LB cells in the presence of IL-21[38]. Naïve B cells isolated from *p38α*[fl/fl]*Vav*[cre] mice, which had a better deletion efficiency in naive B cells than *p38α*[fl/fl]*CD19*[cre] mice (Supplementary Fig. 3a) were used for the cellular experiments. As shown in Fig. 2a, *p38α* deficiency in naive B cells isolated from *p38α*[fl/fl]*Vav*[cre] mice led to diminished iPCs generation and antibody production, but it had no effect on the percentage of iGCB cells. This was consistent with the in vivo observation (Figs. 1a–d). We next sorted iGCB cells, CD138$^-$

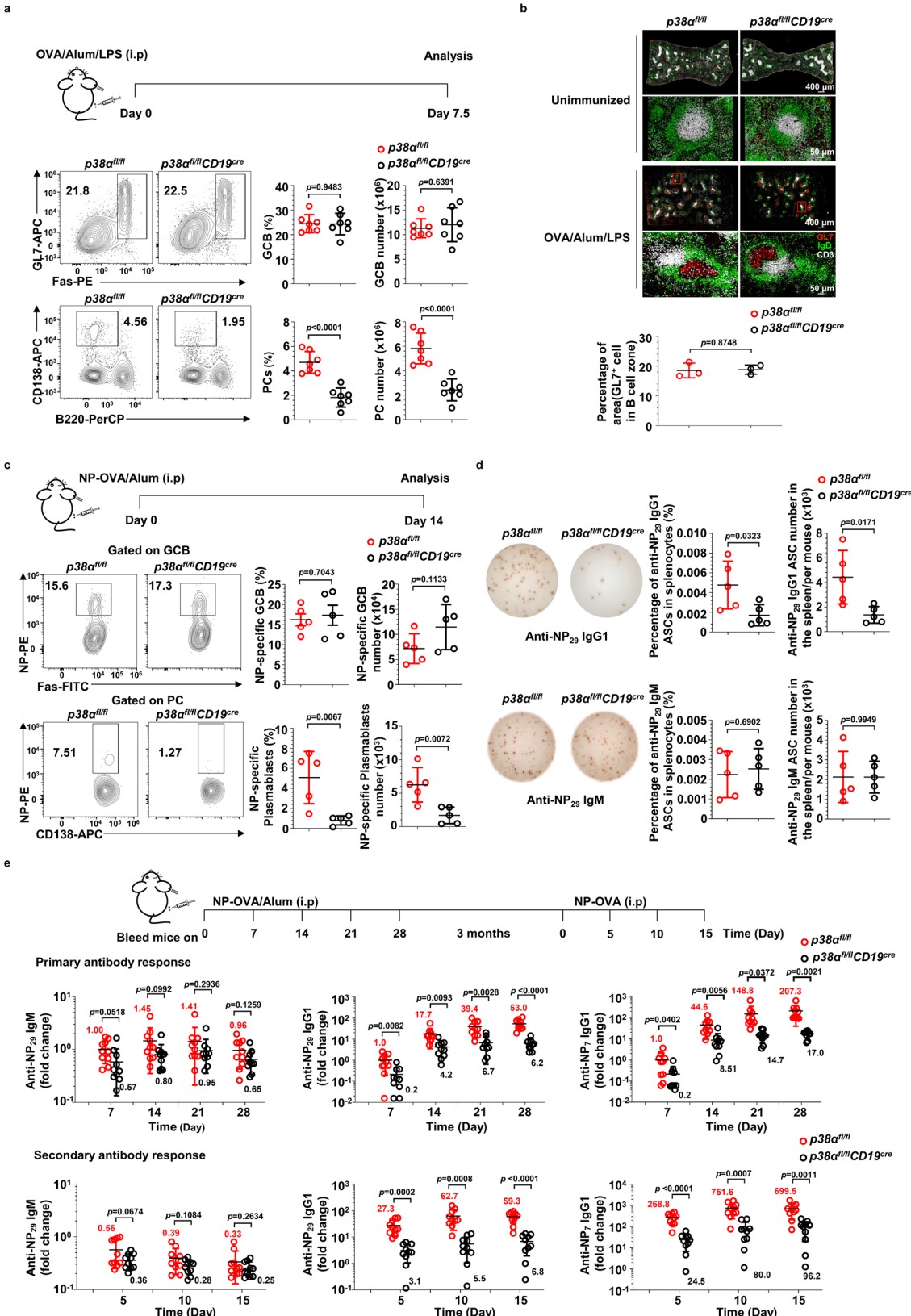

B cells and iPCs (CD19+CD138+) with $p38\alpha^{-/-}$ or WT genotype in the iPC differentiation system and performed RNA-seq analysis. Principal component analysis (PCA) results showed that $p38\alpha^{-/-}$ iPCs and CD138− B cells exhibited very distinct gene expression profiles compared with that of $p38\alpha$ WT iPCs and CD138− B cells, while $p38\alpha^{-/-}$ and WT iGCB cells had similar pattern of gene expression (Fig. 2b). Among the

differentially expressed genes, 864 genes were upregulated and 768 genes were downregulated in $p38\alpha^{-/-}$ iPCs (Fig. 2b) (Supplementary Data. 1). As expected, $p38\alpha$ was detected as one of the most significant differentially expressed genes in both iGCB cell and iPC data sets and the expressions of genes such as *Blimp1, Xbp1, Bip* etc., which are critical for antibody production, were decreased in $p38\alpha$ KO iPCs when

**Fig. 1 | p38α is essential for plasma cell generation and antibody response. a** Flow cytometry analysis of GCB cells (CD3⁻B220⁺Fas⁺GL7⁺) and PCs (B220^low CD138⁺) in the spleen of *p38α^fl/fl CD19^cre* and *p38α^fl/fl* mice (8–10 weeks, *n* = 7 per group) at day 7.5 after immunization (i.p.) with OVA/Alum/LPS. Summary of the percentage and number of GCB cells and PCs. **b** Representative immuno-fluorescence images of germinal centers stained for IgD (naïve B cells, green), GL7 (GCB cells, red) and CD3 (T cells, white) in the spleen of OVA/Alum/LPS-immunized mice (*n* = 3 per group) at day 7.5 post immunization. Unimmunized mice (*n* = 3 per group) were used as controls. Scale bars, 50 or 400 μm. The area occupied by GL7⁺ cells in B cell zone was calculated and shown as percentage of B cell zone.

**c**, **d** Flow cytometry analysis of NP-specific GCB cells (CD3⁻B220⁺Fas⁺GL7⁺NP⁺) and NP-specific plasmablasts (B220^low CD138⁺NP⁺) in the spleen at day 14 post immunization with NP-OVA/Alum (i.p) (*n* = 5 per group) (**c**). NP-specific antibody secreting cells (ASCs) were measured by ELISpot assay (**d**). **e** *p38α^fl/fl CD19^cre* and *p38α^fl/fl* mice were immunized with NP-OVA/Alum (i.p.) (*n* = 10 per group), followed by secondary immunization with NP-OVA (i.p.) at day 120 after primary immunization. Serum was collected at indicated time points, and NP-specific antibody concentration (fold change) was determined by ELISA. Each symbol represents an individual mouse. Small horizontal lines indicate the mean (±s.d.). Data were analyzed by two-tailed unpaired *t*-tests. Source data are provided as a Source Data file.

the RNA-seq data of *p38α* KO and WT cells were compared. To examine the involvement of other p38 family members in PCs generation, we cultured naïve B cells from *p38β^−/−*, *p38γ^−/−*, or *p38δ^−/−* mice in the iPC differentiation system, and found that iPCs generation from those mutant B cells was comparable to their WT counterparts (Supplementary Fig. 3b to 3d). The genetic knockout of *p38β, p38γ,* or *p38δ* in naïve B cells was evidenced by sequencing of p38 isoforms in their corresponding mutation cells (Supplementary Fig. 3e).

To examine whether the effects of p38α on PC generation are cell-intrinsic, purified *p38α^−/−* and WT naive B cells (CD45.2⁺) were co-cultured with CD45.1⁺ naïve WT B cells in a 1:1 ratio (Fig. 2c). The percentage of iPCs derived from *p38α^−/−* B cells was much lower than that of WT B cells, demonstrating a cell-intrinsic role of p38α in iPC generation. The requirement of p38α in PC generation was also examined under several other in vitro culture conditions, including BAFF + CD40L + IL-4, BAFF + CD40L + IL-21, and anti-IgM+IL-4, and iPCs generation from *p38α^−/−* B cells was severely impaired under all those conditions (Supplementary Fig. 3f). We also examined the effect of *p38α* deletion on B cell activation induced by various stimuli, including IL-4, anti-CD40, anti-IgM, and anti-IgM+IL-4, and found no difference (Supplementary Fig. 3g).

The kinase activity is important for the physiological or pathological function of p38α[29,30]. To determine whether the kinase activity of p38α is also important for PC differentiation, we first analyzed p38α activation by an antibody specific for the phosphorylated (active) form of p38α during the differentiation of iGCB cells to iPCs in vitro. IgG heavy (H) chain expression was used as an indicator of iPC differentiation. p38α activation started upon initiation of iPC differentiation, peaked at day 2, and maintained at a high level throughout the course of iPC differentiation (Fig. 2d). We then transduced *p38α^−/−* B cells with retroviruses encoding p38α or two kinase-dead mutants of p38α (K53M or T180A/Y182F). The results showed that replenishment of p38α expression effectively rescued iPC generation, but p38α (K53M) or p38α (T180A/Y182F) failed to do so (Fig. 2e). Thus, kinase activity of p38α is required for p38α to function properly in iPC generation.

### p38α regulates BLIMP1 expression and plasma cell differentiation

To elucidate how p38α regulates PC generation, we further analyzed the effect of *p38α* deficiency on cellular proliferation and differentiation. We labeled iGCB cells with fluorescent dye CFSE and measured mean fluorescence intensity (MFI) of CFSE at different time points during iPC differentiation. Pattern of CFSE dilution was similar in CD19⁺CD138⁺ and CD19⁺CD138⁻ cells (Supplementary Fig. 4a), and total B cell number at day 4 or day 8 showed no significant difference between *p38α^−/−* and WT cells (Supplementary Fig. 4b), indicating that cellular proliferation during iPC differentiation was not altered by *p38α* deletion.

To evaluate the role of p38α in PC differentiation by using BLIMP1 expression level as a simple indicator, we generated *p38α^fl/fl Vav^cre-Blimp1^gfp/+* mice, which express GFP from the endogenous *Blimp1* locus[39]. In these mice, all BLIMP1 positive B cells are antibody-secreting cells and all PCs are BLIMP1 positive[39]. As shown in Fig. 3a, p38α deficiency resulted in decreased frequency of GFP⁺ (BLIMP1⁺) cells.

Importantly, *p38α* deficiency also resulted in decreased BLIMP1 expression (as indicated by GFP expression) in both BLIMP1⁺CD138⁻ cells and BLIMP1⁺CD138⁺ cells (Fig. 3b). The regulation of BLIMP1 expression by p38α was confirmed by immunoblot analysis (Fig. 3c). This result was also supported by immunostaining of intracellular BLIMP1, which showed that the frequency and number of CD138⁺BLIMP1^hi PCs and BLIMP1 expression in CD138⁺BLIMP1^hi PCs were largely decreased in OVA/Alum/LPS-immunized *p38α^fl/fl CD19^cre* mice in comparison with *p38α^fl/fl* mice (Fig. 3d). Thus, our data suggest that p38α promotes BLIMP1 expression and plasma cell differentiation.

To investigate how p38α regulates BLIMP1 expression, we firstly examined mRNA levels of *Blimp1* in cultured *p38α^−/−* and WT B cells during iPC differentiation and found that the induction of *Blimp1* mRNA in *p38α^−/−* B cells was significantly less than WT cells (Fig. 3e), and the less *Blimp1* mRNA was observed in both CD138⁻ and CD138⁺ *p38α^−/−* B cells (Fig. 3f). We next analyzed the stability of *Blimp1* mRNA by treating cells with actinomycin D to block synthesis of new mRNA and found the half-life of *Blimp1* mRNA was comparable in cultured *p38α^−/−* and WT B cells (Supplementary Fig. 4c). Treatment with pro-teasome inhibitor MG132 and protein synthesis inhibitor cyclohex-imide had no effects on BLIMP1 protein stability and translation, respectively, in *p38α^−/−* and WT iPC culture (Supplementary Fig. 4d and 4e). In addition, the kinase activity of p38α was required for *Blimp1* transcription, as ectopic expression of p38α restored the *Blimp1* mRNA level in cultured *p38α^−/−* B cells, but two p38α kinase-dead mutants failed to do so (Fig. 3g). Thus, p38α regulates BLIMP1 expression at the transcription level.

### The p38α-BLIMP1 axis controls iPC differentiation and ER-related transcriptome

To characterize the role of the p38α-BLIMP1 axis in PC differentiation, we examined the effects of p38α replenishment and ectopic BLIMP1 expression on iPCs generation of *p38α^−/−* B cells. As shown in Fig. 4a, b, both p38α and BLIMP1 expression restored iPCs differentiation and IgG1 and IgE production, confirming that BLIMP1 could function downstream of p38α.

To further gain molecular insights into the role of p38α-BLIMP1 axis in PC differentiation, transcriptomes of *p38α^−/−* and WT iPCs, as well as the iPCs differentiated from the *p38α^−/−* cells having trans-duced with retroviruses encoding p38α or BLIMP1, were analyzed by RNA-seq. Differentially expressed genes were classified into 8 groups by whether their expression in *p38α^−/−* iPCs can be restored by ectopic expression of p38α and/or BLIMP1 (Fig. 4c, Supplementary Table 1). Ectopic expression of p38α in cultured *p38α^−/−* B cells restored the expression of most altered genes (majority genes in group1, 2, 5, 6), whereas over-expression of BLIMP1 restored expression of genes in group 5, and to a less extend in group 1. Since the expression of Group1 and Group 5 genes can be restored or partially restored by p38α and BLIMP1 in cultured *p38α^−/−* B cells, these genes should be relevant to iPCs generation and antibody secretion. The Gene Ontology (GO) enrichment analysis showed that the genes in Group 5 were mainly enriched in the processes of cell proliferation and activation (Supplementary Fig. 5a). Since no significant difference in cell division or activation was observed in

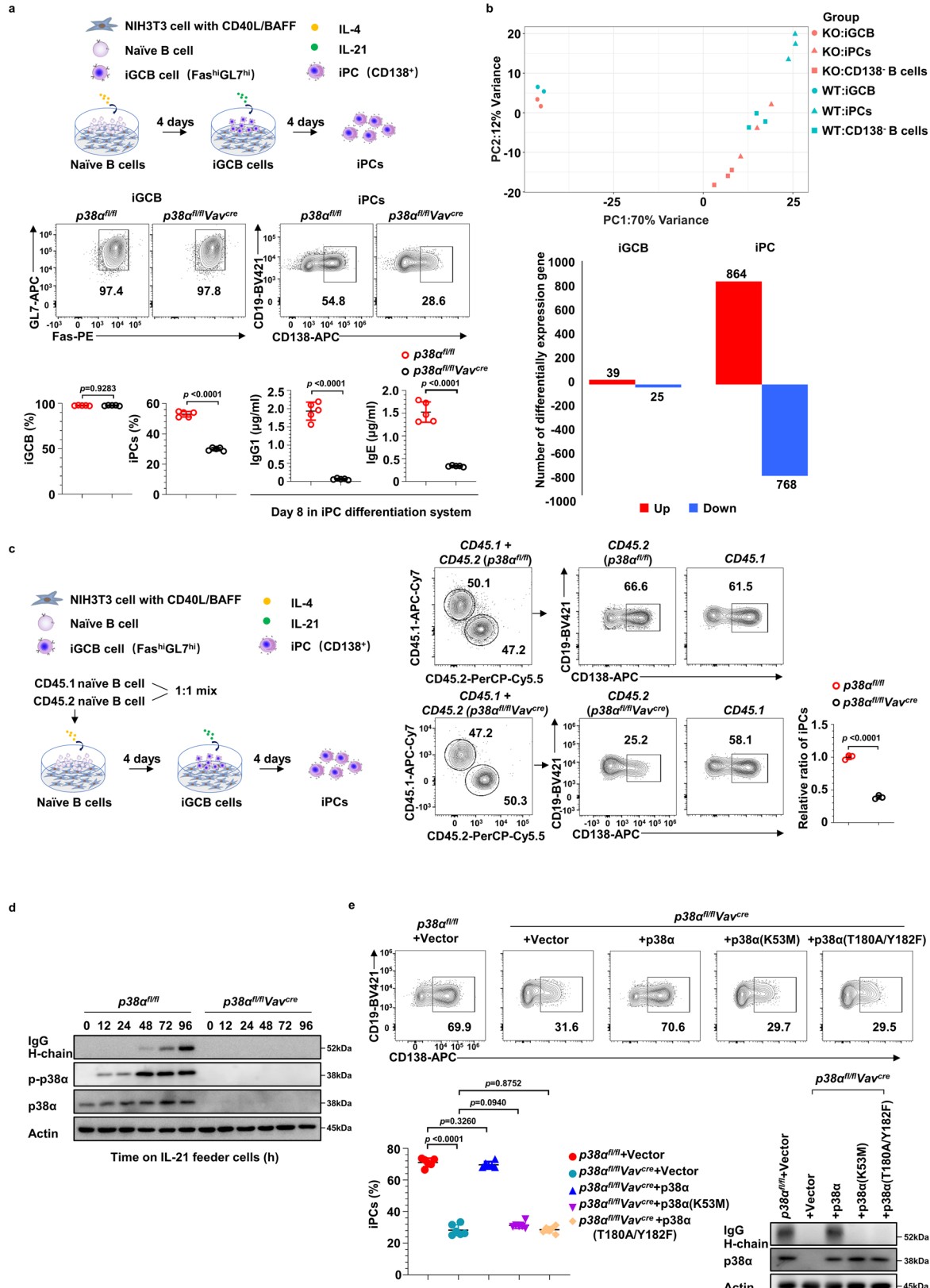

cultured $p38\alpha^{-/-}$ B cells (Supplementary Fig. 3g, 4a and 4b), we did not further evaluate this group of genes. Interestingly, Group1 genes were enriched in the processes of ER stress response (UPR) and immunoglobulin production (Fig. 4d), two key features of PCs. Since expression of either p38α or BLIMP1 was able to restore iPCs generation and antibody secretion in cultured $p38\alpha^{-/-}$ B cells

(Fig. 4a, b), and the BLIMP1 upregulation in $p38\alpha^{-/-}$ cells was less than WT cells, BLIMP1 should be downstream of p38α to regulate ER-related transcriptome in iPC differentiation. In supporting this notion, $p38\alpha^{-/-}$ iPCs exhibited impaired ER expansion and UPR induction, and the $p38\alpha^{-/-}$ iPCs ectopically expressing p38α or BLIMP1 showed normal ER expansion and UPR induction (Fig. 4e, f).

**Fig. 2 | p38α promotes plasma cell generation in a cell-intrinsic and kinase activity-dependent manner. a** Schematic outline of in vitro plasma cell (iPC) differentiation from splenic naïve B cells. Flow cytometry analysis of the percentage of iGCB cells (CD19$^+$Fas$^{hi}$GL7$^{hi}$) at day 4 and iPCs (CD19$^+$CD138$^+$) at day 8 of culture, and supernatant IgG1 and IgE of *p38α* KO and WT B cells at day 8 of iPC culture were measured by ELISA ($n = 6$ per group). **b** Principal-component analysis (PCA) of RNA-seq data from sorted iGCB cells ($n = 2$ per group), iPCs (CD19$^+$CD138$^+$) and CD138$^-$ B cells (CD19$^+$CD138$^-$) of indicated genotypes ($n = 3$ per group). Numbers of upregulated (red) and downregulated (blue) genes (*p* value < 0.01) in *p38α* KO iGCB cells and iPCs in comparison with WT cells. **c** Purified splenic naïve B cells from *p38α*$^{fl/fl}$*Vav*$^{cre}$ or *p38α*$^{fl/fl}$ mice (CD45.2$^+$) were mixed and co-cultured with CD45.1$^+$ WT splenic naïve B cells at a 1:1 ratio. Flow cytometry analysis of iPCs gated on CD45.1$^+$ and CD45.2$^+$ B cells in iPC differentiation system at day 8 of culture ($n = 3$ per

group). **d** Immunoblot analysis of p38α, p-p38α, IgG heavy (H)-chain and Actin in cultured *p38α* KO and WT B cells at indicated time points after being transferred onto IL-21 feeder cells. **e** Cultured *p38α* KO and WT B cells were transduced with retroviruses encoding p38α and p38α kinase-dead mutants (K53M or T180A/Y182F) at day 2 of iGCB cell differentiation. iPCs among retrovirus-transduced B cells (CD19$^+$GFP$^+$CD138$^+$) were analyzed by flow cytometry 6 days after retroviral transduction ($n = 6$ per group). Protein levels of p38α, IgG heavy (H)-chain and Actin in retrovirus-transduced B cells were analyzed by immunoblot. Data were representative of at least three independent experiments (**a**), (**c**–**e**). Each symbol represents a representative sample. Small horizontal lines indicate the mean (±s.d.). Data were analyzed by two-tailed unpaired *t*-tests. Source data are provided as a Source Data file.

## p38α regulates *Blimp1* transcription and PC differentiation via TCF3, TCF4, and IRF4

To elucidate the molecular mechanism underlying p38α regulation of *Blimp1* transcription, we performed a CRISPR/Cas9-mediated screening of p38α substrates in the iPC differentiation system. A total of 136 genes encoding the proteins that were previously reported or predicted as p38α substrates in publications[29,30] or PhosphoSitePlus® PTM website (https://www.cellsignal.com/learn-and-support/phosphositeplus-ptm-database) (Supplementary Data 2), containing protein kinases, transcription factors, transcription regulators, etc., were screened. Three sgRNAs were designed for each gene. Retroviruses encoding each sgRNA (BFP$^+$) were produced individually and utilized to transduce B cells from *Cas9*$^{tg/+}$ mice (GFP$^+$), followed by analysis of the percentage of iPCs among sgRNA-expressing cells (GFP$^+$BFP$^+$) (Fig. 5a). *p38α* was used as a positive control for this system, as shown in Supplementary Fig. 6a, sgRNAs targeting *p38α* dramatically decreased p38α expression and iPCs generation, consistent with results from mouse genetic studies (Fig. 2a). Among the 136 genes screened by this approach, a very few genes were found to be involved in iPC generation and among them the effect of *Tcf3* knockout was similar to that of *p38α* (Fig. 5b, c).

TCF3 is a transcription factor of the E protein family[40], and there is a previously reported TCF3-binding site in the *Blimp1* locus[41,42]. We speculated that TCF3 plays an important role in mediating p38α regulation of PC generation and decided to focus on it in our studies. We confirmed TCF3 binding to this site in WT, but not in cultured *p38α*$^{-/-}$ B cells in the process of iPC differentiation by a CUT&Tag-qPCR assay (Fig. 5d). TCF3 has two isoforms, E12 and E47, resulting from alternative splicing[43]. Ectopic expression of either E12 or E47 in *p38α*$^{-/-}$ B cells led to some increase of BLIMP1 level and partially restored iPC generation (Fig. 5e). Since p38α overexpression has no effects on iPCs generation in cultured *Tcf3*$^{-/-}$ B cells (Fig. 5f), it is consistent with the notion that p38α is upstream of TCF3. The fact that co-deletion of *Tcf3* and *p38α* did not exacerbate the defect in iPC generation caused by *p38α* deletion further supports that p38α and TCF3 function in the same pathway in iPC generation (Fig. 5g). Collectively, our data demonstrated that p38α functions upstream of TCF3 to regulate iPC generation.

It was reported that TCF3 and TCF4 coordinately regulate *Blimp1* transcription during PC differentiation[41,42]. Our results showed that *Tcf4* sgRNA indeed reduced BLIMP1 induction and iPCs generation (Fig. 6a). Moreover, TCF4 binding to the *Blimp1* locus was decreased in cultured *p38α*$^{-/-}$ B cells compared to WT B cells (Fig. 6b). Significantly, ectopic expression of TCF4 partially restored BLIMP1 expression and iPCs generation from cultured *p38α*$^{-/-}$ B cells (Fig. 6c). These results suggest that TCF4 functions downstream of p38α to regulate *Blimp1* transcription and iPC generation.

TCF3 has been shown to cooperate with IRF4, a transcription factor that controls mature B cells positioning, GC reaction, and BLIMP1 expression during PC differentiation[44,45]. We confirmed that *Irf4* sgRNAs inhibited BLIMP1 expression and iPCs generation (Fig. 6d). Diminished IRF4 binding to a previously reported site at the *Blimp1*

locus was observed in *p38α*$^{-/-}$ compared to WT B cells during iGCB cells to iPCs differentiation (Fig. 6e). Furthermore, ectopic IRF4 expression partially restored BLIMP1 induction and iPCs differentiation of cultured *p38α*$^{-/-}$ B cells (Fig. 6f). These results suggest that IRF4 was also involved in p38α-mediated *Blimp1* transcription and iPC generation.

To further examine the functional relationship between p38α, TCF4, and IRF4 in iPC generation, sgRNAs targeting *Tcf4* or *Irf4* were transduced into cultured *p38α*$^{fl/fl}$Cas9$^{tg/+}$ or *p38α*$^{fl/fll}$Vav$^{cre}$Cas9$^{tg/+}$ B cells, followed by examining iPCs differentiation, with *Tcf3*-targeting sgRNAs as positive control. As shown in Fig. 6g, *Tcf4* or *Irf4* deletion did not exacerbate the defect in iPC generation caused by *p38α* deficiency. Moreover, ectopic expression of TCF3, TCF4 and IRF4 partially restored BLIMP1 expression (indicated by GFP expression) in *p38α*$^{fl/fl}$Vav$^{cre}$Blimp1$^{gfp/+}$ B cells (Supplementary Fig. 7a) and IgG1 and IgE secretion in *p38α*$^{fl/fl}$Vav$^{cre}$ B cells (Supplementary Fig. 7b), suggesting that p38α controls BLIMP1 expression and antibody secretion via TCF3, TCF4, and IRF4.

It needs to note that *p38α* deletion did not affect the protein levels of TCF3 and TCF4 and slightly decreased the IRF4 protein level (Supplementary Fig. 7c and 7d). The reduction of IRF4 level in *p38α*$^{-/-}$ B cells is most likely resulted from the decrease of BLIMP1 level in *p38α*$^{-/-}$ B cells (Fig. 3c) since BLIMP1 and IRF4 could positively upregulate each other during PC differentiation[12,46]. This notion is also supported by the observation of the decrease of IRF4 protein level in *Blimp1*$^{-/-}$ B cells in iPC culture (Supplementary Fig. 7e). TCF3, TCF4 and IRF4 should function downstream of p38α in PC differentiation as deletion of *Tcf3*, *Tcf4*, and *Irf4* had no additive effect on *p38α* knockout (Figs. 5g and 6g). Taken together, our findings demonstrated that TCF3, TCF4 and IRF4 were downstream of p38α to promote *Blimp1* transcription and iPC differentiation.

MEF2C is a known p38α substrate that also functions in Tregs to sustain the BLIMP1-activated transcriptional program[47,48]. MEF2C certainly participated in PC generation since it was identified as the top 3 hits in our screen for the genes required for iPC generation (Fig. 5b). We thus evaluated the possible role of the p38α-MEF2C axis in PC differentiation. We showed that *p38α* knockout by CRISPR/Cas9 in cultured B cells in iPC differentiation system had no markable effect on B cell number, different from *Mef2c* knockout that significantly decreased B cell number (Supplementary Fig. 8a). Decrease of B cell viability in germinal center was reported in a study using *Mef2c*$^{fl/fl}$CD19$^{cre}$ mice[33], suggesting that the marked reduction of B cell number by *Mef2c* knockout may be due to cell death of cultured B cells. Ectopic expression of MEF2C, or its phospho-mimetic mutants MEF2C-3E (T293E/T300E/S387E) or MEF2C-3D (T293D/T300D/S387D) in cultured *p38α*$^{-/-}$ B cells had no rescue effect on *p38α* knockout caused defect in iPC differentiation (Supplementary Fig. 8b). Moreover, CRISPR/Cas9-mediated knockout of *Mef2c* in both *p38α*$^{-/-}$ and WT iGCB cells decreased PC differentiation in the iPC differentiation system, suggesting additive effects of MEF2C and p38α in this process (Supplementary Fig. 8c). These data showed that MEF2C indeed plays a

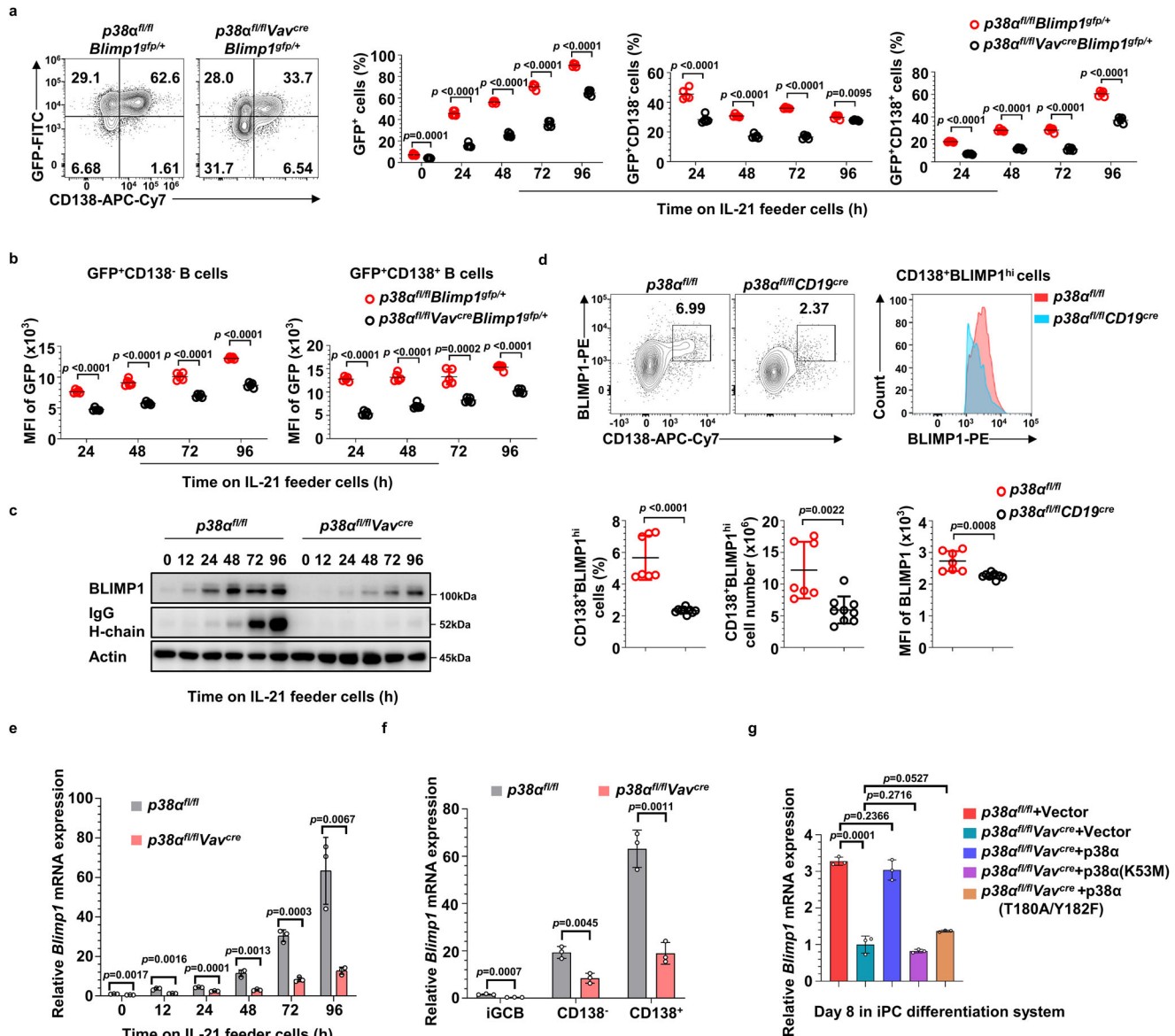

**Fig. 3 | p38α regulates BLIMP1 expression. a** Flow cytometry analysis of BLIMP1-expressing cells (CD19⁺GFP⁺), BLIMP1-expressing CD138⁻ cells (CD19⁺GFP⁺CD138⁻) and BLIMP1-expressing CD138⁺ (CD19⁺GFP⁺CD138⁺) cells differentiated from *p38α^fl/fl^Blimp1^gfp/+^* and *p38α^fl/fl^Vav^cre^Blimp1^gfp/+^* B cells in iPC culture (n = 5 per group). **b** MFI of GFP in BLIMP1-expressing CD138⁻ B cells (CD19⁺GFP⁺CD138⁻) and BLIMP1-expressing CD138⁺ B cells (CD19⁺GFP⁺CD138⁺) from (**a**) were determined by flow cytometry (n = 5 per group). **c** Immunoblot analysis of BLIMP1, IgG H-chain and Actin in cultured B cells at indicated time points after *p38α* KO and WT iGCB cells transferred onto IL-21 feeder cells. **d** Flow cytometry analysis of BLIMP1 expression by intracellular staining for BLIMP1 in splenic B cells from OVA/Alum/LPS-immunized mice described as Fig. 1a (n ≥ 7 per group). **e** Quantitative RT-PCR analysis of *Blimp1* mRNA in cultured *p38α* KO and WT B cells at indicated time points after

transferred onto IL-21 feeder cells (n = 3 per group). **f** *p38α* KO and WT iGCB cells at day 4, CD138⁻ B cells (CD19⁺CD138⁻) and CD138⁺ B cells (CD19⁺CD138⁺) at day 8 were sorted, mRNA level of *Blimp1* in these subsets were analyzed by quantitative RT-PCR (n = 3 per group). **g** Cultured *p38α* KO B cells were transduced with retroviruses encoding p38α or p38α kinase-dead mutants (K53M or T180A/Y182F) and transferred onto IL-21 feeder cells. *Blimp1* mRNA levels were measured by quantitative RT-PCR after 4 days of culture (n = 3 per group). Data were representative of at least three independent experiments. Each symbol represents a representative sample or an individual mouse (**d**). Small horizontal lines indicate the mean (±s.d.). Data were analyzed by two-tailed unpaired *t*-tests. Source data are provided as a Source Data file.

role in iPC differentiation but this role is different from that of p38α in iPC differentiation, and the activation of MEF2C by p38α is unlikely to be a mechanism in iPC differentiation.

### The activation of TCF3, TCF4, and IRF4 downstream of p38α in PC differentiation

Previously study reported that p38α phosphorylates TCF3(E47) at Ser140 and promotes MyoD/E47 association and muscle-specific gene transcription[49]. However, ectopic expression of a E47 S140D or S140E phosphomimetic mutant in cultured *p38α^−/−^* B cells restored iPC

differentiation to a degree similar to WT TCF3 (Supplementary Fig. 9a), suggesting that phosphorylation of Ser140 is dispensable for the function of E47 in iPC generation.

To investigate the mechanism underlying how p38α regulates TCF3, TCF4, and IRF4 activity, we firstly tested whether the kinase activity of p38α is required for transcriptional activation of TCF3, TCF4, and IRF4 during iGCB to iPC differentiation. We introduced luciferase reporter genes containing TCF3, TCF4, or IRF4 binding site from *Blimp1* respectively into the in vitro iPC differentiation system, and observed increased expression of all of the three reporters in

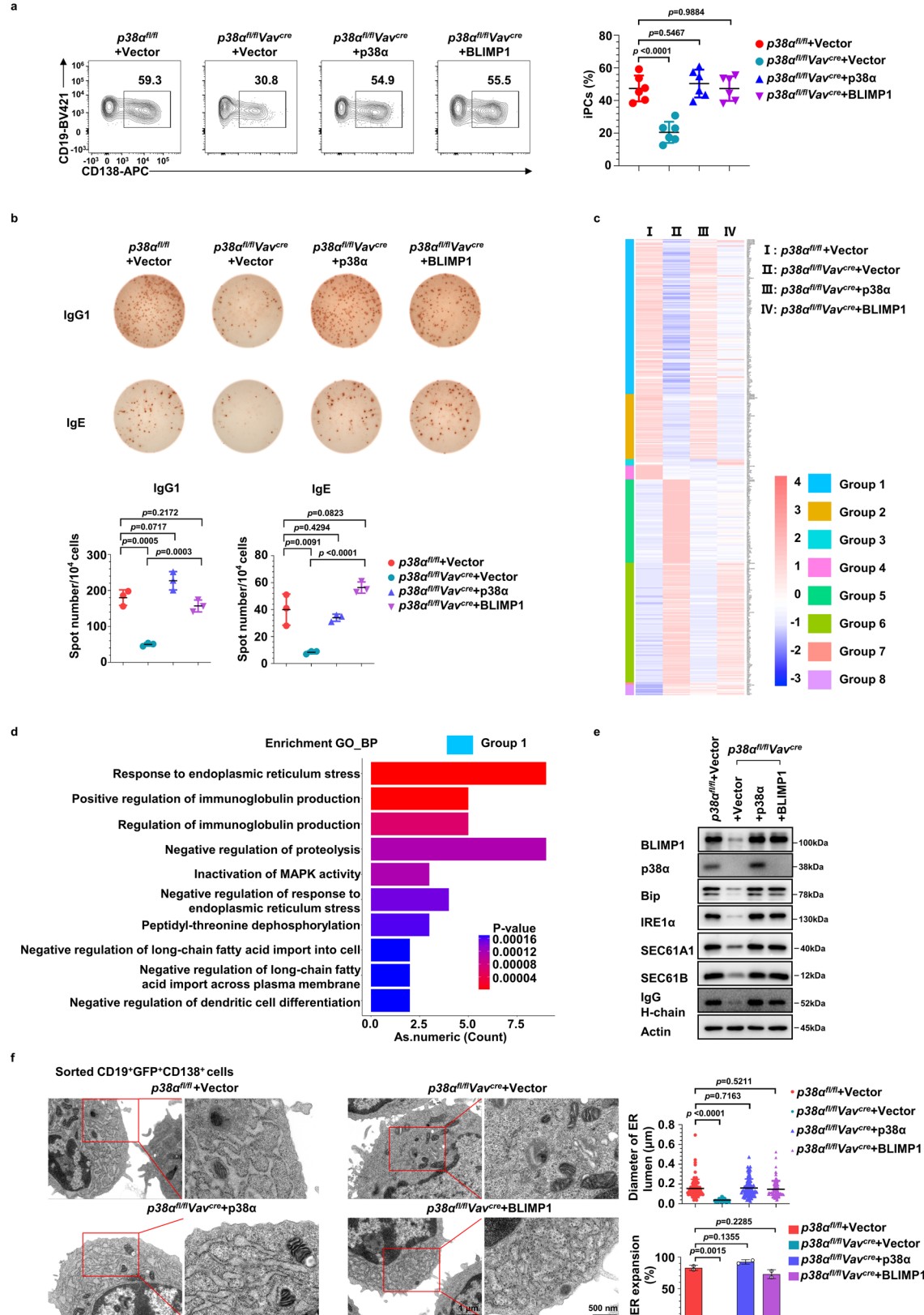

cultured *p38α* WT but not *p38α*$^{-/-}$ B cells. And ectopic expression of WT but not the kinase-dead mutant of p38α in cultured *p38α*$^{-/-}$ B cells restored the upregulation of these luciferase reporters (Fig. 7a). Thus, the kinase activity of p38α is required for the downstream activation of TCF3, TCF4, and IRF4 in iPC differentiation.

To identify putative phosphorylation sites on TCF3 (E47), TCF4, and IRF4, in vitro kinase assays were performed using recombinant MKK6E (a p38α activator) and p38α as kinases, and TCF3 (E47), TCF4, or IRF4 proteins as substrates, the phosphorylation sites were analyzed by mass spectrometry (MS). 7 S/T residues on IRF4, 7 S residues on E47,

**Fig. 4 | The p38α-BLIMP1 axis controls iPC differentiation and ER-related transcriptome. a** Cultured *p38α* KO and WT B cells were transduced with retroviruses encoding p38α or BLIMP1 at day 2 of in vitro differentiation. Flow cytometry analysis of iPCs in retrovirally transduced B cells (CD19⁺GFP⁺CD138⁺) 6 days after transduction (*n* = 6 per group). **b** ELISpot analysis of IgG1 and IgE antibody-secreting cells among cultured B cells as (**a**) (*n* = 3 per group). **c** Heatmap analysis of differential expression genes (foldchange > 2) in iPCs of indicated genotypes. Group 1: decreased in *p38α* KO iPCs, restored by p38α or BLIMP1 overexpression. Group 2: decreased in *p38α* KO iPCs, restored by p38α but not BLIMP1 overexpression. Group 3: decreased in *p38α* KO iPCs, restored by BLIMP1 but not p38α overexpression. Group 4: decreased in *p38α* KO iPCs, restored by neither p38α nor BLIMP1 overexpression. Group 5: increased in *p38α* KO iPCs, restored by p38α or BLIMP1 overexpression. Group 6: increased in *p38α* KO iPCs, restored by p38α but not BLIMP1 overexpression. Group 7: increased in *p38α* KO iPCs, restored by BLIMP1 but not p38α overexpression. Group 8: increased in *p38α* KO iPCs, restored neither p38α nor BLIMP1 overexpression. **d** Gene Ontology enrichment analysis of Group 1 from (**c**). **e** Immunoblot analysis of BLIMP1, p38α, IgG H-chain, Actin and UPR-related proteins, Bip, IRE1α, SEC61A1 and SEC61B in retrovirally transduced B cells at day 8 in iPC culture. **f** Cultured *p38α* KO and WT B cells were transduced with retroviruses encoding p38α or BLIMP1 at day 2 of in vitro differentiation. Representative transmission electron microscopy images of sorted iPCs in retrovirally transduced B cells (CD19⁺GFP⁺CD138⁺) of indicated groups at day 8. Scale bars, 500 nm. Percentage of endoplasmic reticulum (ER) expansion and diameter of ER lumen in iPCs of indicated groups (*n* ≥ 100 per group). Data were representative of at least three independent experiments (**a**, **b**, **e**, **f**). Each symbol represents a representative sample. Small horizontal lines indicate the mean (±s.d.). Data were analyzed by one-side hypergeometric test (**d**) and two-tailed unpaired *t*-tests (**a**, **f**). Source data are provided as a Source Data file.

and 4 S/T residues on TCF4 were identified being phosphorylated by p38α in vitro (Supplementary Data. 3).

μE2-μE5 reporter was commonly used in the field to indicate TCF3 and TCF4 dependent gene expression[50] and IFNβ reporter which indicates IRF4-mediated gene expression[51,52] (termed as E-box reporter and IRF4 reporter respectively in this study). These reporters could also signpost transactivation of TCF3/TCF4 and IRF4 in the in vitro iPC differentiation system (Supplementary Fig. 9b). Because E-box reporter and IRF4 reporter had strong signals, we used them in the reporter co-expression assay to reveal which p38α phosphorylation sites could be involved in the transcription activation of TCF3 (E47), TCF4, and IRF4. *p38*⁻/⁻ 293 A cell line, which lacks the expression of all four p38 family members was used for co-expression[53]. Co-expression of IRF4 with IRF4 reporter enhanced reporter gene expression and activation of p38α dramatically enhanced IRF4-dependent reporter expression (Supplementary Fig. 9c). We mutated each of the potential phosphorylation sites in IRF4 (Supplementary Fig. 9d) and expressed them in the co-expression assay. T267A, T268A and T269A mutation, but not the mutation on other sites rendered IRF4 no long response to p38α in increasing IRF4 reporter gene expression (Fig. 7b, Supplementary Fig. 9e) and the triple T to A mutant (T267A/T268A/T269A) of IRF4 completely lost the transcription activity. The role of IRF4 phosphorylation on T267, T268 and T269 sites was further demonstrated by the data that these single T to A mutants of IRF4 failed to restore BLIMP1 expression and iPC generation in cultured *Irf4*⁻/⁻ B cells (Fig. 7c). Thus, our results demonstrated that IRF4 phosphorylation at T267, T268 and T269 by p38α is the activation mechanism of IRF4 in PC differentiation.

Using similar approaches, we studied activation mechanisms of TCF3 and TCF4. All the E47 mutants harboring one or more putative phosphorylation sites of p38α being mutated to alanine responded to p38α activation normally in term of E-box reporter expression. In addition, E47 mutants with one or all 13 potential MAP kinase-phosphorylation site(s) (S/T that precede a Proline residue) mutated to A or with several multiple site mutations also showed no defect in response to p38α activation. Similar results were also observed in studying the regulation of TCF4 by p38α. Till now we do not know the mechanisms of p38α-dependent activation of TCF3 and TCF4.

### The p38α-BLIMP1 axis regulates PC generation from in vivo generated GCB cells

iGCB cells generated in the 40LB culture system do not fully mimic GCB cells generated in vivo[38,54]. We therefore isolated splenic GCB cells from OVA/Alum/LPS-immunized *p38α*^fl/fl^CD19^cre^ and *p38α*^fl/fl^ mice and cultured them on 40LB cells with IL-21 to differentiate them into iPCs. As shown in Fig. 8a, iPCs differentiation from *p38α*⁻/⁻ GCB cells were significantly decreased compared with iPCs differentiation from WT GCB cells. Furthermore, kinase activity of p38α was also required for iPC generation, as ectopic expression of p38α fully restored iPCs differentiation of *p38α*⁻/⁻ GCB cells, but the two kinase-dead mutants of p38α failed to do so (Fig. 8b).

We next ectopically expressed BLIMP1, TCF3, TCF4 and IRF4 in sorted *p38α*⁻/⁻ GCB cells, followed by iPC differentiation on 40LB cells with IL-21. As shown in Fig. 8c, ectopic expression of BLIMP1 fully restored iPC generation from *p38α*⁻/⁻ GCB cells. Moreover, ectopic expression of TCF3, TCF4 and IRF4 partially restored iPC generation from *p38α*⁻/⁻ GCB cells. Taken together, these results confirmed the p38α-BLIMP1 axis regulates iPC differentiation from in vivo generated GCB cells.

### The p38α-BLIMP1 axis regulates iPC generation from LPS-activated B cells

As mentioned before, p38α was required for NP-LPS-induced PC generation in vivo (Supplementary Fig. 2a), we therefore investigated whether the p38α-BLIMP1 axis was also involved in this process. As shown in Fig. 9a, *p38α* deletion reduced LPS-induced cell differentiation to iPCs with no effect on the total number of cells (in vitro induced PCs (iPCs) + CD138⁻ B cells). And ectopic expression the two kinase-dead mutants of p38α failed to restore iPCs generations from cultured *p38α*⁻/⁻ B cells, suggesting the kinase activity of p38α was also required for LPS-induced iPC differentiation (Fig. 9b). The ratio of BLIMP1-expressing cells (GFP⁺) and the expression level of BLIMP1 (indicated by MFI of GFP) in BLIMP1⁺ cells were also significantly decreased in LPS-treated *p38α*⁻/⁻ B cells (Fig. 9c). Moreover, ectopic expression of p38α or BLIMP1 fully restored iPC generation of LPS-treated *p38α*⁻/⁻ B cells, while TCF3, TCF4 or IRF4 over-expression only partially restored it (Fig. 9d). Thus, the p38α-BLIMP1 axis also regulates iPC generation from LPS-activated B cells, and TCF3, TCF4 and IRF4 appears to also function downstream of p38α in this process.

## Discussion

p38α is a member of the p38 MAPK family and plays critical roles in the differentiation of many types of cells[29,30]. Here, we show that p38α controls PC differentiation by upregulating *Blimp1* transcription via TCF3, TCF4, and IRF4. p38α can directly activate IRF4 by phosphorylating it on T267, T268, and T269 (Supplementary Fig. 9d). The S270 C-terminal side to these three threonine sites in IRF4 was also phosphorylated by p38α in vitro. Since S270 precedes a proline in IRF4, S270 is a typical candidate site for MAP kinase phosphorylation. Although its A mutation had no effect on p38α-mediated enhancement of IRF4 transcription activity, S270 is likely to be phosphorylated in vivo (Fig. 7b). Since BLIMP1 also regulates IRF4 expression[12], p38α-mediated IRF4 activation should enhance the BLIMP1 and IRF4 positive feedback loop in PC differentiation. However, we unfortunately have not elucidated the mechanisms for the activation of TCF3 and TCF4 by p38α in PC differentiation. Whether TCF3 and TCF4 are direct substrates of p38α in the process of PC differentiation and whether they have partners in their transactivation etc. await further investigation.

TCF3 (E47) is previously identified as a p38α substrate[49]. Both TCF4 and IRF4 are previously shown to interact or cooperate with TCF3 and function as partners for TCF3[41,42,44]. As *Blimp1* harbors several

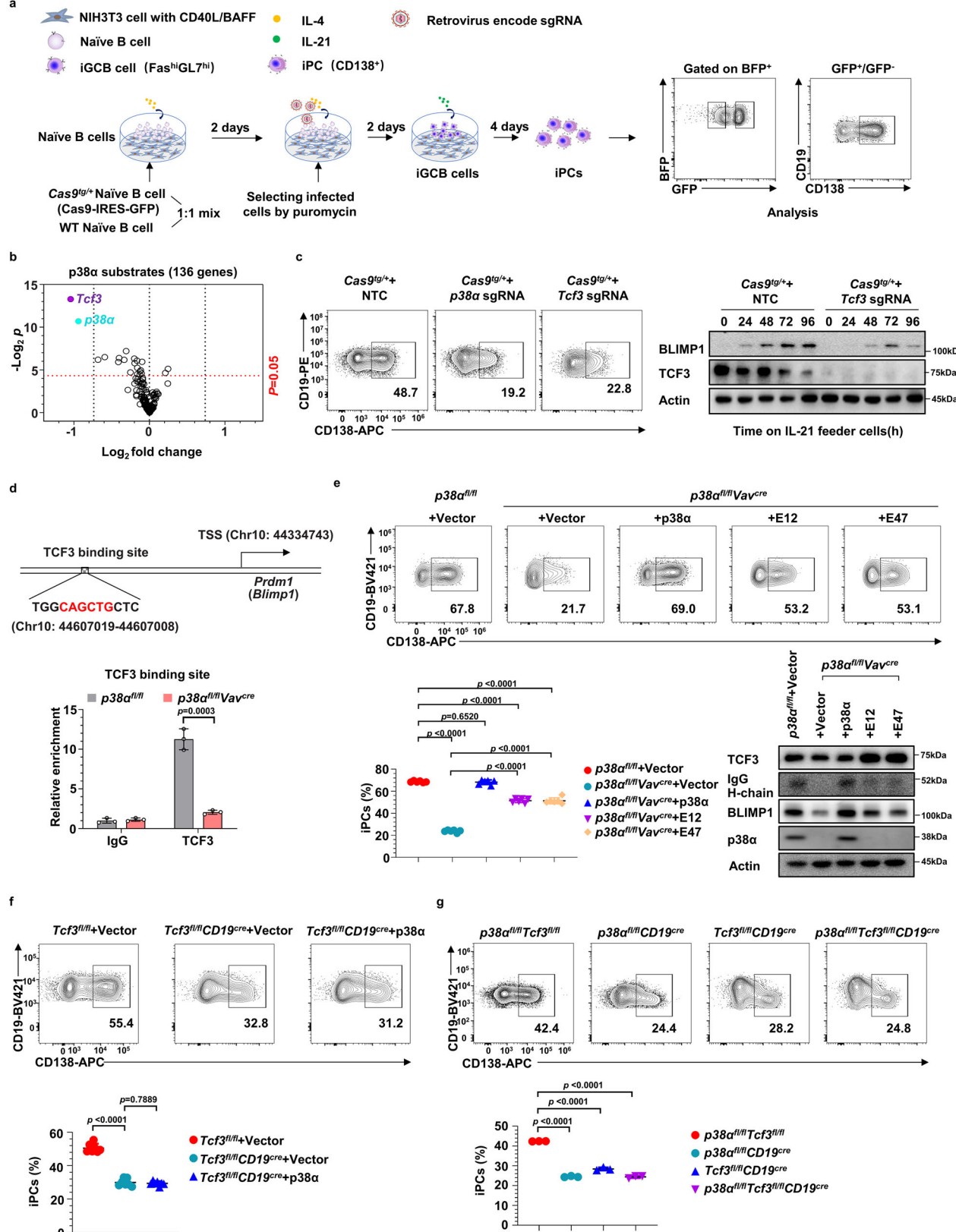

transcription initiation sites and multiple regulatory elements for its tightly regulated transcription[55], it is conceivable that multiple transcription factors were engaged in the upregulation of *Blimp1* by p38α. p38α was reported to phosphorylate TCF3 (E47) at Ser140[49]. However, S140D or S140E mutant had no difference from WT TCF3 (E47) in restoring iPC differentiation of cultured *p38α*[−/−] B cells (Supplementary

Fig. 9a), indicating that the previously described regulation mechanism of TCF3 (E47) by p38α is not applicable to iPC differentiation.

The p38α-MEF2C signaling axis has been implicated to be involved in the development, proliferation, and function of B cells[32,33]. p38 inhibitor inhibited MEF2C-EBF1 mediated lymphoid/myeloid fate decision toward B cell in vitro[31] and BCR-stimulated B cell proliferation

**Fig. 5 | p38α controls *Blimp1* transcription through TCF3. a** Schematic outline of CRISPR/Cas9 screen of p38α substrates in iPC differentiation. Naive B cells from *Cas9*$^{tg/+}$ mice (GFP$^+$) and WT littermates (GFP$^-$) were mixed at a 1:1 ratio and cultured. Cultured B cells were transduced with retroviruses encoding individual sgRNAs targeting p38α substrates, BFP, and puromycin-resistance gene at day 2. iPCs at day 8 among retrovirus-transduced Cas9-expressing B cells (CD19$^+$GFP$^+$BFP$^+$CD138$^+$) were analyzed by flow cytometry. **b** Summary of the screen results in (**a**). *X* axis: The ratio between iPCs percentages of p38α substrates sgRNA-expressing and NTC sgRNA-expressing B cells (Fold change); *Y* axis: The variation of effect on iPCs generation of 3 sgRNAs targeting gene of interest as measured by Students' *t*-tests. **c** Flow cytometry analysis of iPCs among cultured *Cas9*$^{tg/+}$ B cells transduced with retroviruses encoding *p38α* or *Tcf3* sgRNA (CD19$^+$GFP$^+$BFP$^+$CD138$^+$). Immunoblot analysis of BLIMP1, TCF3 and Actin expression in *Tcf3* sgRNA expressing B cells at indicated time of culture. **d** Schematic diagram of a putative TCF3-binding site in the *Prdm1* locus. *p38α* KO and WT B cells cultured on IL-21 feeder cells for

2 days were utilized for CUT&Tag assay and immunoprecipitated with anti-TCF3 antibody. The enrichment of TCF3 binding genomic fragments containing the reported TCF3 binding site in *Blimp1* enhancer region was measured by quantitative RT-PCR (*n* = 3 per group). **e** Flow cytometry analysis of iPCs among cultured *p38α* KO B cells transduced with retroviruses encoding the TCF3 splicing isoforms E12 and E47 (CD19$^+$GFP$^+$CD138$^+$) (*n* = 6 per group), protein levels of TCF3, IgG H-chain, BLIMP1, p38α, and Actin were analyzed by immunoblot. **f** Flow cytometry analysis of iPCs among cultured *Tcf3* KO B cells transduced with retroviruses encoding p38α (CD19$^+$GFP$^+$CD138$^+$) (*n* = 7 per group). **g** Flow cytometry analysis of iPCs (CD19$^+$CD138$^+$) among cultured *p38α*$^{fl/fl}$*CD19*$^{cre}$, *Tcf3*$^{fl/fl}$*CD19*$^{cre}$, *p38α*$^{fl/fl}$*Tcf3*$^{fl/fl}$*CD19*$^{cre}$, *p38α*$^{fl/fl}$*Tcf3*$^{fl/fl}$ B cells at day 8 of in vitro iPC differentiation (*n* = 3 per group). NTC, non-targeting control. Data were representative of at least three independent experiments (**b–g**). Each symbol represents a representative sample. Small horizontal lines indicate the mean (±s.d.). Data were analyzed by two-tailed unpaired *t*-tests. Source data are provided as a Source Data file.

for which MEF2C is crucial[32]. However, *p38α*$^{-/-}$ chimeric mice had normal B cell development, activation and proliferation[35], and *Mef2c*$^{fl/fl}$*CD19*$^{cre}$ mice were normal in B cell development but defective in GCB cell formation[32,33]. In agreement with the data of *p38α*$^{-/-}$ chimeric mice, our studies showed that *p38α* deletion does not affect B cell development, activation and division (Supplementary Fig. 1a and 1b, Supplementary Figs. 3g, 4a and 4b). Different from the impaired GCB generation in *Mef2c*$^{fl/fl}$*CD19*$^{cre}$ mice, genetic deletion of *p38α* did not have a significant effect on GCB cell generation (Fig. 1a and c). Gene expression profiles of *p38α*$^{-/-}$ and WT iGCB cells were almost identical with 64 differentially expressed genes (Fig. 2b) (Supplementary Data 1). These data suggested that the role of MEF2C in GCB cell formation may not be regulated by p38α. The role of p38α and MEF2C in PC differentiation has not been specifically studied previously. We showed in this study that the effect of *p38α* knockout on B cell survival was different from that of *Mef2c* knockout (Supplementary Fig. 8a). MEF2C, or its phospho-mimetic mutants cannot rescue the defect in iPC differentiation caused by *p38α* knockout (Supplementary Fig. 8b). And the effect of *Mef2c* knockout and *p38α* knockout had additive effect in impairing iPC differentiation (Supplementary Fig. 8c). Collectively, these data suggest that at least part of the functions of p38α and MEF2C are not in the same signaling axis during B cell generation, GCB cell formation and PC differentiation.

It was reported that the basal antibody concentration in serum was normal in *p38α*$^{-/-}$ chimeric mice compared with their WT littermates[35]. However, serum IgG1 antibody concentration was found to be decreased in *p38α*$^{fl/fl}$*CD19*$^{cre}$ mice in our study. This discrepancy might be due to the big variation in the limited number of mice (*n* = 3) used in the previous study[35]. Reduced TD antibody response was also observed in *p38γ*$^{-/-}$*p38δ*$^{-/-}$ mice, but B cell-intrinsic expression of p38γ and p38δ contributes little to this phenotype[56]. We showed that deletion of *p38γ* and *p38δ* did not affect iPC differentiation (Supplementary Fig. 3c and 3d). The published data and ours on *p38γ* and *p38δ* did not conflict.

In addition, the p38α-BLIMP1 axis is also involved in LPS-induced PC generation. The shared usage of the p38α-BLIMP1 pathway in the process of GCB cell to PC differentiation and a type I but not type II T cell-independent antigen-induced PC differentiation is interesting and worthwhile to be further investigated in the future.

Elevated expression and activation of p38α are observed in several autoantibody-driven autoimmune diseases, such as SLE and RA[57,58]. Treatment with p38 inhibitors alleviated symptoms and prolonged the survival of MRL-lpr mice, a widely used SLE mouse model, but the mechanism of action has been poorly understood[59,60]. As depositions of IgG and anti-dsDNA auto-antibodies are ameliorated in MRL-lpr mice treated with p38 inhibitors, and elevated levels of auto-antibodies cause organ and tissue damage in MRL-lpr mice, it is most likely that p38 inhibitors inhibit PC differentiation in this model.

Therefore, targeting p38α specifically in B cells could be a therapeutic strategy to treat those autoimmune diseases to avoid clinical toxicity of p38 inhibitors.

## Methods

### Mice

*p38α*$^{fl/fl}$ (Jax stock, 031129) was previously described[61]. *p38β*$^{-/-}$, *p38γ*$^{-/-}$, *p38δ*$^{-/-}$ mice were a kindly gift from Huiping Jiang in Boehringer Ingelheim Pharmaceuticals, Inc. *Vav*$^{cre}$ (Jax stock, 008610), *CD19*$^{cre}$ (Jax stock, 006785), C57BL/6J (Jax stock, 000664), *Tcf3*$^{fl/fl}$ (isolated from Jax stock, 024511), *Irf4*$^{fl/fl}$ (Jax stock, 009380), *Blimp1*$^{fl/fl}$ (Jax stock, 008100), *CD45.1* B6.SJL (Jax stock, 002014), *Cas9*$^{tg/+}$ (Jax stock, 026179) mice were purchased from the Jackson Laboratory. *Blimp1*$^{gfp/+}$ mice were a kind gift from Stephen L. Nutt[39].

All the mice were back-crossed to the C57BL/6J background for at least six generations, and housed in a specific pathogen-free facility under a 12 h light-dark cycle at the Xiamen University Laboratory Animal Center. The light time was from 8 a.m. to 8 p.m., and the room temperature were kept at 22–24 °C and humidity at 50–70%. Unless stated otherwise (5–6 weeks old mice used for analysis B cells development), both male and female mice (8–10 weeks old) were used in this study. The experimental/control animals were co-housed, and mice were euthanized by a carbon dioxide method. All mouse experiments were approved by the Animal Care and Use Committee of Xiamen University (XMULAC20180073).

### Splenic naïve B cell purification

Spleen was grinded gently and single-cell suspension was collected by filtering with 70 μM filters. RBCs were depleted with ammonium chloride lysis buffer (Beyotime, C3702) and naïve B cells were isolated with negative selection by using biotinylated monoclonal Abs (mAbs) against CD43 (Biolegend, 121204), CD5 (Biolegend, 100604), CD9 (BD Biosciences, 558749), CD93 (eBioscience, 13-5892-85), Ter-119 (Biolegend, 116204), Streptavidin Particles Plus-DM (BD Biosciences, 557812) and cultured with RPMI 1640 medium.

### In vitro plasma cell (iPC) differentiation system

The iPC differentiation system was previously described[38]. Briefly, naive B cells purified from mouse spleen were cultured on NIH3T3 feeder cells expressing CD40L, BAFF and IL-4 (termed IL-4 feeder cells) on six-well plates at a density of $1 \times 10^5$ cells/well for 4 days for differentiation into iGCB cells, which were subsequently transferred to fresh NIH3T3 feeder cells expressing CD40L, BAFF, and IL-21 (termed IL-21 feeder cells) at a density of $1 \times 10^5$ cells/well and cultured for another 4 days for differentiation into iPCs. iPCs were collected at indicated time for analysis. Feeder cells were irradiated with a dose of 120 Gy in a RS-2000 irradiator (Rad Source) before use.

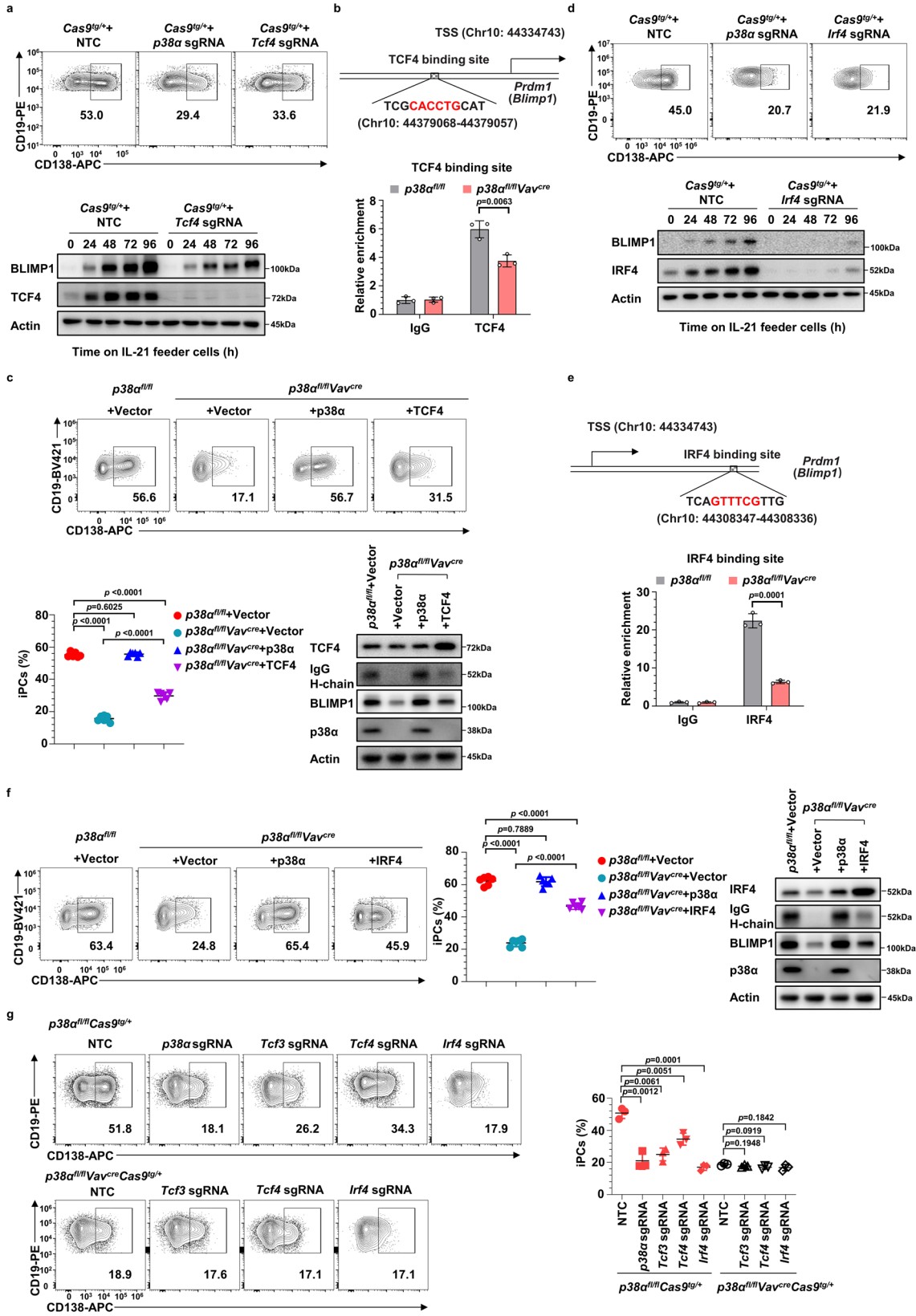

For the iPCs differentiated from in vivo GCB cells experiments, splenic GCB cells were sorted from OVA/Alum/LPS immunized mice at day 7.5, and cultured on IL-21 feeder cells for 4 days for differentiation into iPCs. iPCs were collected at indicated time for analysis.

**Other in vitro plasma cell differentiation systems**
Purified naïve B cells were plated at a density of $1 \times 10^5$ cells/ml on a six-well plate with NIH3T3 feeder cells expressing CD40L and BAFF (termed 40LB cells). IL-4 (10 ng/ml) or IL-21(10 ng/ml) was added to medium. Cells were cultured for 8 days (IL-4) or 6 days

**Fig. 6 | TCF4 and IRF4 function downstream of p38α to regulate *Blimp1* transcription. a** Flow cytometry analysis of iPCs differentiated from cultured *Cas9*^(tg/+) B cells transduced with retroviruses encoding *p38α* or *Tcf4* sgRNA (CD19⁺GFP⁺BFP⁺ CD138⁺). Immunoblot analysis of BLIMP1, TCF4, and Actin expression in *Tcf4* sgRNA expressing B cells at indicated time points after being transferred onto IL-21 feeder cells. **b** Schematic diagram of a putative TCF4-binding site in the *Prdm1* locus. CUT&Tag analysis of TCF4 binding genomic fragments including the reported TCF4 binding site in *Blimp1* enhancer region in cultured *p38α* KO and WT B cells at day 2 after transferred onto IL-21 feeder cells (*n* = 3 per group). **c** Flow cytometry analysis of iPCs among cultured *p38α* KO B cells transduced with retroviruses encoding TCF4 (CD19⁺GFP⁺CD138⁺) (*n* = 6 per group), protein levels of TCF4, IgG

H-chain, BLIMP1, p38α and Actin were analyzed by immunoblot. **d**–**f** Same as described in (**a**–**c**) except that *Irf4* sgRNAs were used or IRF4 was overexpressed in cultured *p38α* KO B cells. **g** Cultured *p38α*^(fl/fl)*Vav*^(cre)*Cas9*^(tg/+) and *p38α*^(fl/fl)*Cas9*^(tg/+) B cells were transduced with retroviruses encoding sgRNA targeting *p38α*, *Tcf3*, *Tcf4* and *Irf4* at day 2 of culture (*n* = 3 per group). Percentages of iPCs in sgRNA transduced B cells (CD19⁺GFP⁺BFP⁺CD138⁺) were analyzed by flow cytometry at day 8. NTC, non-targeting control. Data were representative of at least three independent experiments. Each symbol represents a representative sample. Small horizontal lines indicate the mean (±s.d.). Data were analyzed by two-tailed unpaired *t*-tests. Source data are provided as a Source Data file.

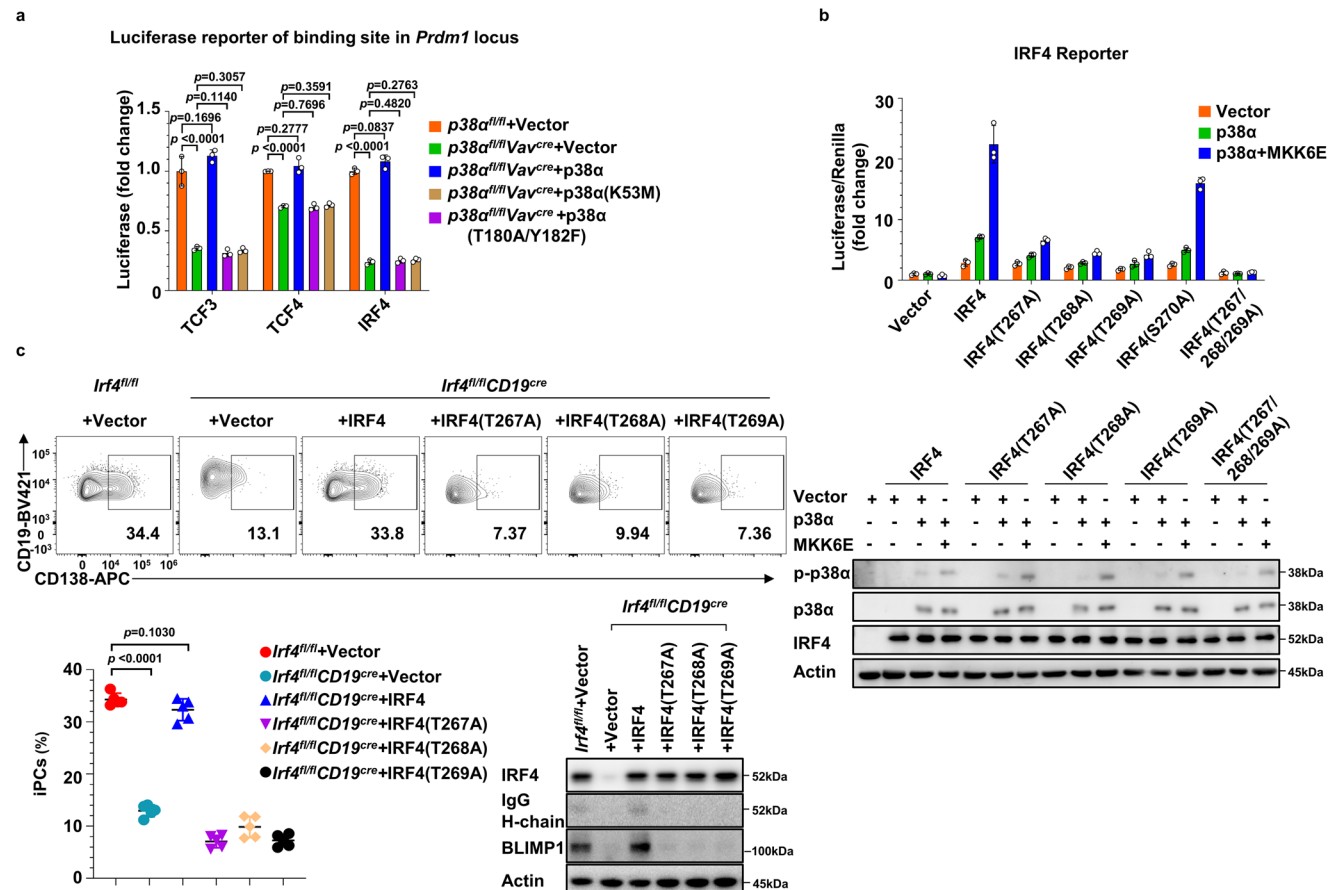

**Fig. 7 | The activation of TCF3, TCF4 and IRF4 downstream of p38α in PC differentiation. a** Cultured *p38α*^(−/−) B cells were transduced with retroviruses encoding p38α or p38α kinase-dead mutations (K53M or T180A/Y182F) (GFP⁺) and luciferase reporter harboring TCF3, TCF4 or IRF4 binding sites in *Prdm1* enhance region at day 2, and luciferase activity in retrovirus-infected B cells (CD19⁺GFP⁺) was analyzed at day 8 in iPC culture (*n* = 3 per group). **b** *p38*^(−/−) 293 A cells were transfected with indicated plasmids, and the luciferase activity of IRF4 mutants was analyzed 24 h after transfection. Immunoblot analysis of p38α, p-p38α, IRF4 and Actin in *p38*^(−/−) 293 A cells transfected with indicated plasmids (n = 3 per group).

**c** Cultured *Irf4*^(−/−) B cells were transduced with retroviruses encoding IRF4 or IRF4 mutants (T267A, T268A, T269A) at day 2, iPCs in retrovirally transduced B cells (CD19⁺GFP⁺CD138⁺) 6 days after transduction were analyzed by flow cytometry (*n* = 5 per group), and protein levels of IRF4, IgG H-chain, BLIMP1 and Actin were analyzed by immunoblot. Data were representative of at least three independent experiments. Each symbol represents a representative sample. Small horizontal lines indicate the mean (±s.d.). Data were analyzed by two-tailed unpaired t-tests. Source data are provided as a Source Data file.

(IL-21), followed by flow cytometry analysis. For anti-IgM and IL-4 stimulation, purified naive B cells were plated at a density of $2 \times 10^5$ cells/ml on a six-well plate, stimulated with anti-IgM (1 μg/ml) + IL-4 (10 ng/ml) for 4 days, and analyzed by flow cytometry.

### Plasmids
Full-length and mutant cDNA of p38α was PCR-amplified from plasmids previously reported and cloned into the Rv-IRES-GFP vector by a ligase-independent clone method. *Blimp1, Tcf3*

(E12/E47), *Tcf4,* and *Irf4* cDNAs were PCR-amplified from mouse iPCs cDNA generated by reverse transcription and cloned into the Rv-IRES-GFP vector. Mutant *Irf4* and *Tcf3* plasmids were constructed by a bridging PCR method and cloned into the Rv-IRES-GFP vector respectively. For sgRNA-expressing plasmids, oligos were cloned into the pMSCV-IRES-BFP vector by T4 DNA ligase. All sgRNA sequences used in this study are listed in Supplementary Data 2. All the constructs were verified by Sanger sequencing.

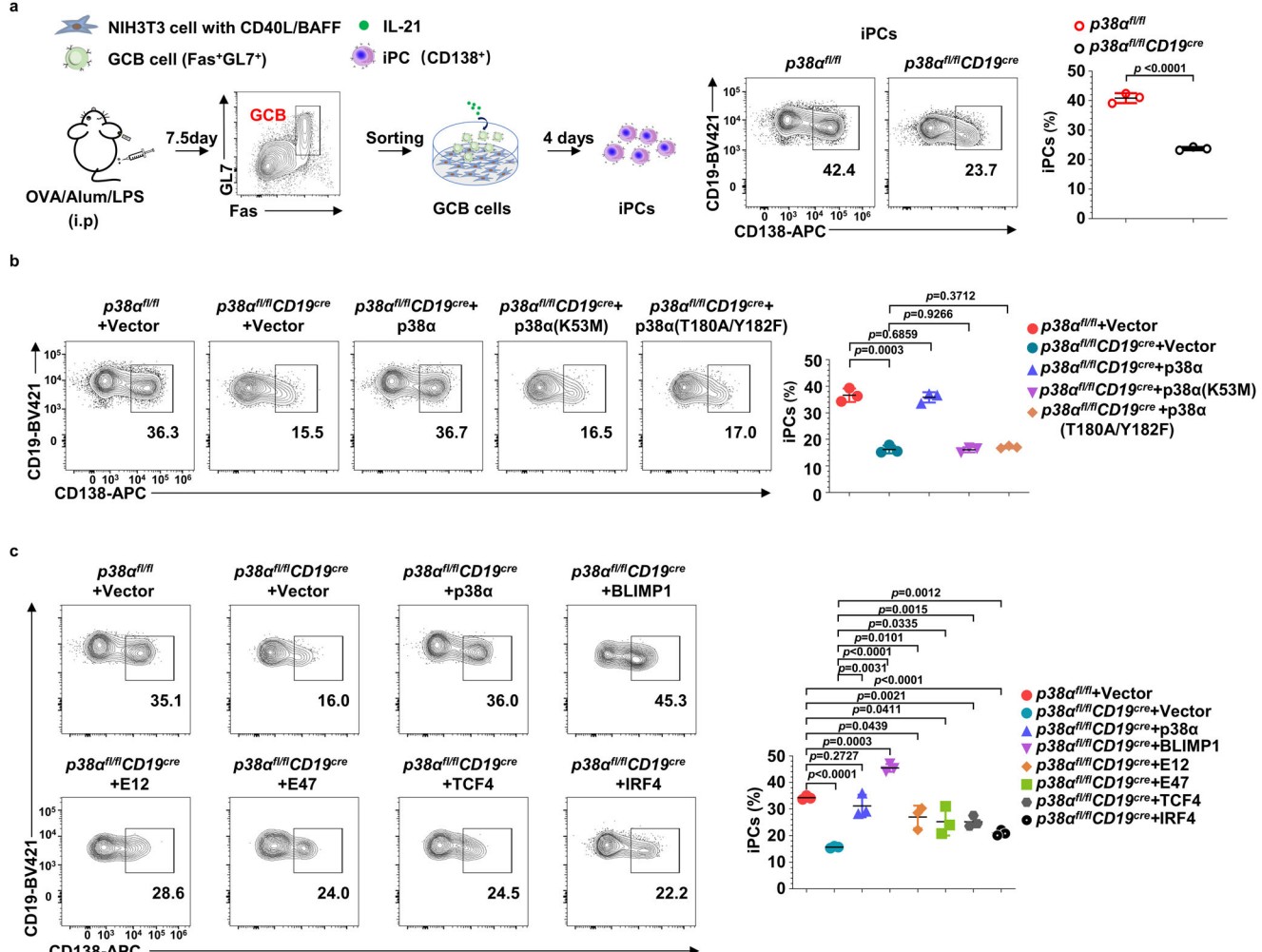

**Fig. 8 | The p38α-BLIMP1 axis controls PC generation differentiated from in vivo generated GCB cells. a** Sort GCB cells (CD3⁻B220⁺Fas⁺GL7⁺) from the spleen of OVA/Alum/LPS immunized mice (*n* = 10 per group) at day 7.5 were mixed and cultured on IL-21 feeder cells, and iPCs differentiated from *p38α* KO and WT GCB cells were analyzed at day 4. **b** Sorted *p38α* KO GCB cells from the spleen of OVA/Alum/LPS immunized mice (*n* = 10 per group) were mixed and cultured on IL-21 feeder cells and transduced with retroviruses encoding p38α or p38α kinase-dead mutants (K53M or T180A/Y182F) 1 day later. The percentage of iPCs in retrovirus-transduced B cells (CD19⁺GFP⁺CD138⁺) was analyzed by flow cytometry 3 days after transduction. **c** Flow cytometry analysis of iPCs among cultured *p38α* KO GCB cells from the spleen of OVA/Alum/LPS immunized mice (*n* = 10 per group) transduced with retroviruses encoding p38α, BLIMP1, TCF3 (E12 or E47), TCF4, or IRF4 (CD19⁺GFP⁺CD138⁺). Data were representative of at least three independent experiments. Each symbol represents a representative sample. Small horizontal lines indicate the mean (±s.d.). Data were analyzed by two-tailed unpaired t-tests. Source data are provided as a Source Data file.

## Retroviral transduction

For retrovirus packaging, indicated retroviral constructs together with retroviral packaging plasmid (pCL-Eco) were transfected into 293T cells by calcium phosphate transfection. Retrovirus-containing supernatants were collected at 48 h after transfection. Naive B cells cultured on a six-well plate in the iPC differentiation system at day 2 were transduced with 2 ml/well retrovirus-containing supernatants with 5 μg/ml polybrene and centrifuged at 2500 r.p.m. for 30 min at 37 °C. Culture medium was replaced with fresh medium 6–8 h after infection.

## Flow cytometry analysis

For cell surface marker staining, cells were incubated with antibodies in 1× FACS buffer (1% FBS in PBS) for 30 min at 4 °C, and washed with 1× FACS buffer twice before analysis. For intracellular staining, surface marker-stained cells were fixed with Fixation/Permeabilization buffer (eBioscience, 00-5523-00) for 10 min at room temperature, washed with 1× Permeabilization buffer, and treated with indicated antibodies following the manufacturer's instructions. All the stained samples were

kept at 4 °C for indicated analysis. Flow cytometry data were analyzed with the FlowJo software (version: 10.6.2).

## Immunoblot or Western blot

Sorted B cells were re-suspended in lysis buffer with protease inhibitor cocktail (Sigma, S8820-2TAB). Total cell lysates were sonicated and centrifuged at 20,000 × *g* for 30 min at 4 °C and an equal volume of 2× SDS sample buffer was added to the supernatants. Cell lysate samples were resolved on SDS-PAGE gels and transferred to PVDF membrane (Millipore, IPVH00010), and the membrane was incubated with primary antibody overnight at 4 °C. After being washed three times in 1× TBST buffer, horseradish peroxidase (HRP)-conjugated goat anti-rabbit, goat anti-mouse or goat anti-rat antibody was incubated with the membrane for 1 h at room temperature. The membrane was then washed three times with 1× TBST buffer. Protein bands were visualized with ECL Western Blotting Detection Reagent (Ncmbio, P10300) and images were acquired with Amersham Imager 600 (GE Healthcare) following the manufacturer's instructions. Intensities of blots were quantified with Image J software (version:

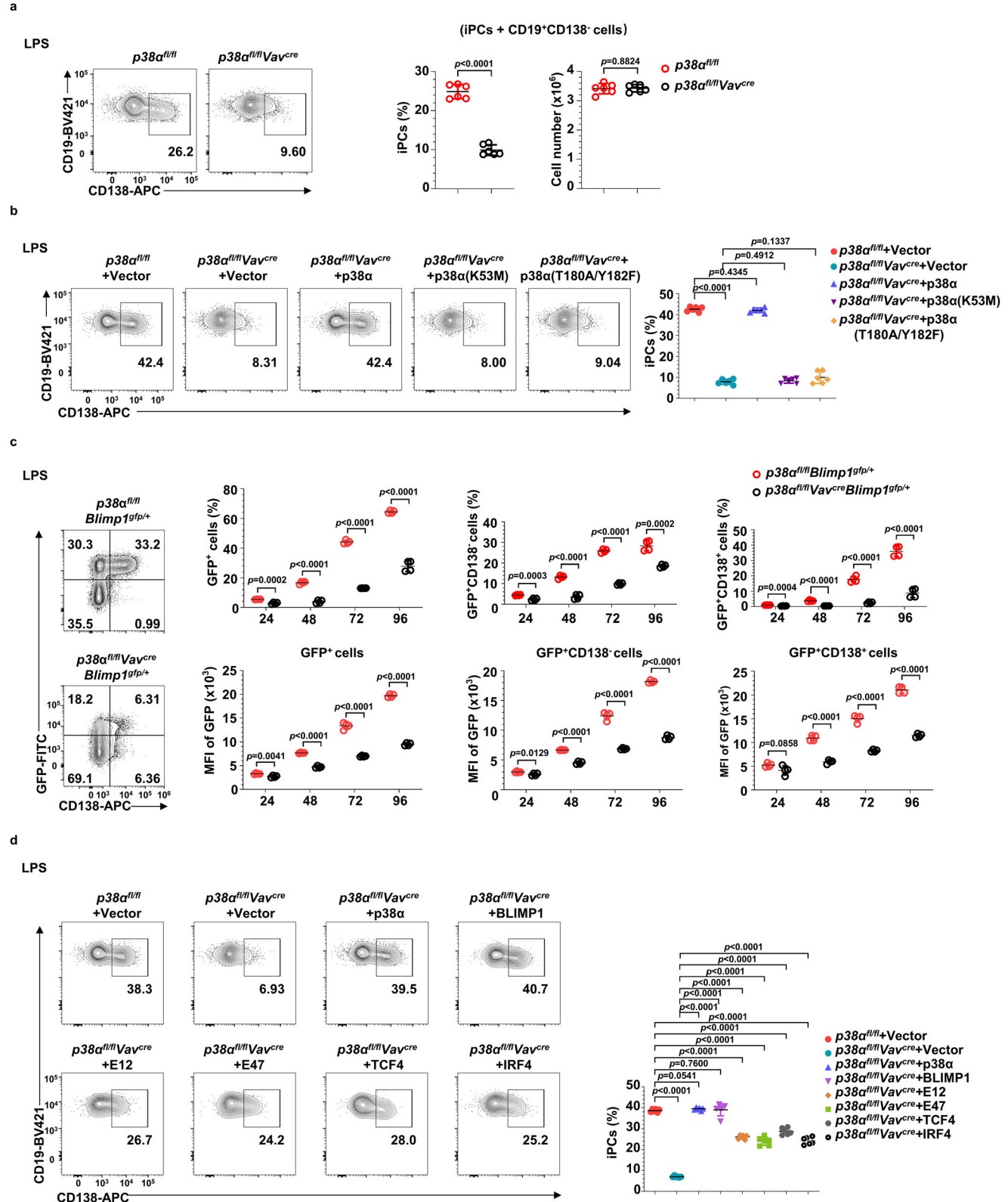

1.8.0). Uncropped and unprocessed scans of the blots are in the Source Data file.

## Antibodies and reagents

The following antibodies were from Biolegend: APC anti-mouse CD21 (123412, 1:200), APC anti-mouse CD117 (105812, 1:200), APC anti-mouse GL7 (144617, 1:200), APC-Cyanine7 anti-mouse CD69 (104525,

1:200), biotin anti-CD5 (100604, 1:200), biotin anti-CD43 (121204, 1:200), biotin anti-Ter119 (116204, 1:200), FITC anti-mouse CD43 (143204, 1:200), eFluor 450 anti-mouse B220 (103227, 1:200), PE anti-mouse CD23 (101608, 1:200), and PE anti-mouse CD25 (102008, 1:200).

The following antibodies and reagents were from eBioscience: APC anti-mouse CD19 (17-0193-82, 1:200), APC-eFluor™ 780 anti-

**Fig. 9 | The p38α-BLIMP1 signal axis is required for LPS-induced PC generation.** **a** Naive B cells from the spleen of *p38α^fl/fl^Vav^cre^* or *p38α^fl/fl^* mice were stimulated with LPS for 4 days (*n* = 6 per group). The percentage of iPCs (CD19⁺CD138⁺) was analyzed by flow cytometry. **b** Cultured *p38α* KO and WT B cells were transduced with retroviruses encoding p38α or p38α kinase-dead mutants (K53M or T180A/Y182F) at day 1 of LPS-induced iPCs generation. iPCs among retrovirus-infected B cells (CD19⁺GFP⁺CD138⁺) were analyzed by flow cytometry 3 days after retroviral transduction (*n* = 6 per group). **c** Flow cytometry analysis of BLIMP1-expressing cells (CD19⁺GFP⁺), BLIMP1-expressing CD138⁻ B cells (CD19⁺GFP⁺CD138⁻) and BLIMP1-expressing iPCs (CD19⁺GFP⁺CD138⁺) differentiated from cultured

*p38α^fl/fl^Vav^cre^Blimp1^gfp/+^* and *p38α^fl/fl^Blimp1^gfp/+^* B cells stimulated with LPS and MFI of GFP (indicating Blimp1 expression) in BLIMP1-expressing cells, GFP⁺CD138⁻ B cells and iPCs (*n* = 4 per group). **d** Cultured *p38α* KO and WT B cells were transduced with retroviruses encoding BLIMP1, TCF3 (E12 or E47), TCF4, or IRF4 at day 1 of LPS-induced iPCs generation. iPCs among retrovirus-infected B cells (CD19⁺GFP⁺CD138⁺) were analyzed by flow cytometry 3 days after retroviral transduction (*n* = 6 per group). Data were representative of at least three independent experiments. Each symbol represents a representative sample. Small horizontal lines indicate the mean (±s.d.). Data were analyzed by two-tailed unpaired *t*-tests. Source data are provided as a Source Data file.

mouse IgM (47-5790-80, 1:200), anti-CD40 (16-0402-86, 1:200), biotin anti-mouse CD93 (13-5892-85, 1:200), eFluor 450 anti-mouse CD19 (48-0193-82, 1:200), PE anti-mouse CD5 (12-0051-82, 1:200), PE anti-mouse CD86 (12-0862-83, 1:200), PE-Cyanine7 anti-mouse CD3e (25-0031-82, 1:200), PE-Cyanine7 anti-mouse CD80 (25-0801-82, 1:200), PerCP-CY5.5 anti-mouse B220 (45-0452-82, 1:200), streptavidin APC-eFluor™ 780 (47-4317-82, 1:200), Propidium Iodide (BMS500PI) and Foxp3/transcription factor staining buffer set (00-5523).

The following antibodies and reagents were from BD Pharmingen™: APC-CY7 anti-mouse CD45.1 (560579, 1:200) and Cytofix/Cytoperm™ fixation/permeablization kit (554714).

The following antibodies and reagents were from BD Biosciences: Biotin anti-mouse CD9 (558749, 1:200), APC anti-mouse CD138 (558626, 1:200), biotin anti-mouse CD138 (553713, 1:100), FITC anti-mouse CD95 (561979, 1:200), PE anti-mouse BLIMP1 (564702, 1:100), PerCP-CY5.5 anti-mouse CD45.2 (552950, 1:200) and streptavidin particles Plus-DM (557812).

The following antibodies were from Southern Biotech: Mouse IgE-UNLB (0114-01, 1:50-1:1,000,000), IgG1-UNLB (0102-01, 1:50-1:1,000,000), IgG3-UNLB (0105-01, 1:50-1:1,000,000), IgM-UNLB (0101-01, 1:50-1:1,000,000), goat anti-mouse IgE-BIOT (1110-08, 1:500), goat anti-mouse IgG1-BIOT (1070-08, 1:500), goat anti-mouse IgG3-BIOT (1100-08, 1:500), goat anti-mouse IgM-BIOT (1020-08, 1:500), goat anti-mouse Kappa-UNLB (1050-01, 1:200) and goat anti-mouse Kappa-UNLB (1060-01, 1:200).

The following antibodies were from CST: Anti-p38α (9228S, 1:1000) and anti-phospho-p38 (Thy180/Tyr182) (9211S, 1:500).

The following antibodies were from Proteintech: Anti-TCF3 (21242-1-AP, 1:1000), anti-IRF4 (11247-2-AP, 1:1000) and anti-TCF4 (22337-1-AP, 1:1000).

The following antibodies were from Santa Cruz: Anti-Actin (sc-47778, 1:1000) and anti-BLIMP1 (sc-47732, 1:1000).

The following reagents were from Sigma: Cycloheximide (C7698), 4-nitrophenyl phosphate (N2765) and Protease inhibitor cocktail (S8820-2TAB).

Anti-GAPDH (AC002, 1:1000) was from ABclonal. Anti-IgM (MCA1293, 1:1000) was from AbD Serotec. Recombine murine IL-4 (AF-214-14) was from PeproTech. TransStart® top green qPCR supermix (AQ131), TransScript® one-step gDNA removal and cDNA synthesis supermix kit (AT311) were from Transgen. CFSE (C34554) and TRIZOL (15596026) were from Invitrogen. MG132 (S2619) was from Selleck. NP-BSA (N5050-10) was from Biosearch Technologies. AEC peroxidase (HRP) substrate kit (SK-4200) and Av-HRP (A-2004) were from Vector Laboratories.

## CUT&Tag-qPCR

Cultured B cells on the IL-21 feeder cells at day 2 were sorted by flow cytometry. $1 \times 10^5$ cells were used for CUT&Tag (Vazyme, TD901) experiment following the manufacturer's recommendations. Briefly, nuclei were isolated and thawed, mixed with activated Concanavalin A beads and magnetized to remove the liquid with a pipettor and resuspended in 1× Wash buffer. Nuclei were then incubated with primary antibody (TCF3 antibody, Santa Cruz sc-416; TCF4 antibody, Santa Cruz sc-393407; IRF4 antibody, Santa Cruz sc-48338, the

corresponding rabbit polyclonal IgG CST 3900 S was used as control antibody) for 2 h and secondary antibody for 1 h in 1× Wash buffer. The beads were washed and resuspended in pG-Tn5 in 300-Wash buffer (Wash buffer containing 300 mM NaCl) for 1 h. After one wash with 300-Wash buffer, samples were incubated in 300-Wash buffer supplemented with 10 mM MgCl₂ for 1 h at 37 °C to fragment chromatin. Reactions were stopped by addition of SDS to 0.16% and protease K to 0.3 mg/mL, and incubated at 58 °C for 1 h. DNA was purified by phenol:chloroform extraction and ethanol precipitation. Libraries were prepared with 14 cycles of PCR with 10 s combined annealing and extension for enrichment of short DNA fragments. DNA fragments were amplified by qPCR primers designed to cover the reported *Blimp1* enhancer region for indicated transcription factors.

qPCR primers used in this study were:

*Tcf3*- cut tag -F: 5′ - CTGAACACCTGGAAGGGCT -3′;
*Tcf3*- cut tag -R: 5′ - GAAGGTGATTGCTCAATGTTCA-3′;
*Tcf4*- cut tag -F: 5′ - CTGAGGTCCTTCCCAAGC-3′;
*Tcf4*- cut tag -R: 5′- AAGAGTGACGTGGCACAGG-3′;
*Irf4*- cut tag -F: 5′ - AGTTTTCTACTTGACTGTTCAGTTTC -3′;
*Irf4*- cut tag -R: 5′ - ACATCATCTGTGCTGGAGGC-3′;

## Quantitative RT-PCR

Total RNA isolated from sorted B cells was extracted with TRIZOL reagent (Invitrogen, 15596026). The TransScript® One-Step gDNA Removal and cDNA Synthesis SuperMix kit (Transgen, AT311) were used to prepare cDNA. Quantitative real-time PCR was performed with the TransStart® Top Green qPCR SuperMix (Transgen, AQ131) and a CFX96 Real-Time System (Bio-Rad, C1000) using standard PCR conditions. Gene expression was normalized with β-actin and data were presented as fold difference of the normalized values between indicated samples and WT B cells by the '2-ΔΔCT' (change in cycling threshold) method. RT-PCR data were analyzed using Bio-Rad CFX Manager software (version: 1.0.1035.131).

Primers used in this study were:

*Blimp1*-RT-F: 5′ -TGCGGAGAGGCTCCACTA-3′;
*Blimp1*-RT-R: 5′ -TGGGTTGCTTTCCGTTTG-3′;
*β-actin*-RT-F: 5′ -CTAAGGCCAACCGTGAAAAG-3′;
*β-actin*-RT-R: 5′ -ACCAGAGGCATACAGGGACA-3′.

## ELISpot and ELISA

NP-specific ASCs in spleen or ASCs in the iPC differentiation system were detected using an ELISpot assay on a MultiScreen 96-well filtration plate (Millipore, MSIPS4W10) coated with 10 μg/ml NP₂₉-BSA or anti-mouse kappa-UNLB and anti-mouse lambda-UNLB (2.5 μg/ml). Serially diluted cells were added to individual wells in triplicate and incubated for 6 h at 37 °C with 5% CO₂. Anti-NP IgG1, Anti-IgG1 or Anti-IgE spots were revealed by biotin-conjugated anti-mouse IgG1 or IgE Ab (Southern Biotechnology, 1070-08) in conjunction with Av-HRP and AEC substrate (Vector Laboratories, A-2004& SK-4200).

For ELISA, flat-bottom 96-well plates (NUNC, 439454) were coated with NP-BSA (10 μg/ml) or anti-mouse kappa-UNLB and anti-mouse lambda-UNLB (2.5 μg/ml) in PBS. Nonspecific binding was blocked with 0.5% BSA in PBS. Serum samples were serially diluted in 0.5% BSA in

PBS and were incubated in blocked plates at room temperature for 2 h. Plates were incubated with biotin-conjugated anti-Ig (Southern Biotech) for 2 h and with streptavidin–alkaline phosphatase for 1 h (Roche, 1089161), and then incubated with alkaline phosphatase substrate solution containing 4-nitro-phenyl phosphate (Sigma-Aldrich, N2765) for color development, followed by quantification on Molecular Devices (VERSA max).

### Luciferase reporter assay

Previously described TCF3, TCF4 or IRF4 binding sites in the *Blimp1* enhancer region[15,41,42] were cloned into a retrovirus luciferase reporter vector modified from the pMSCV vector. The E-box and IRF4 luciferase reporter plasmid were reported previously[50–52]. B cells in the iPC differentiation system were transduced with luciferase reporter plasmids by retrovirus infection. After being transferred to IL-21 feeder cells, luciferase activities of $4 \times 10^5$ B cells were measured with the Luciferase Assay System (Promega, E1960) at indicated time points. Luciferase activities were normalized with day 0 values. Indicated plasmids are transduced into $p38^{-/-}$ 293 A cells through a PEI method, and luciferase activity were measured 24 h after transduction.

The sequence of TCF3, TCF4 and IRF4 binding sites in Blimp1 enhance region are:

TCF3 binding sequence: AAGAGCAGCTGCCAGC
TCF4 binding sequence: CGCACCTGC
IRF4 binding sequence: CACAACGAAACTGAAC

### PEI transfection

293 A cells with a 90% confluent in 24-well plates were replaced with fresh DMEM before PEI transfection. Plasmids (500 ng) and PEI (2 μl, 1 μg/μl) were individually incubated with 25 μl Opti-MEM (Invitrogen, 31985070). These two reagents were subsequently mixed together briefly for 15 min at room temperature and then added into 293 A cells, and the culture media was replaced with fresh DMEM 8 h after transfection.

### His-tag protein purification

CDSs of related genes were cloned into pET-28a vector harboring an N-terminal His-tag. The constructs were verified by DNA sequencing and transformed into competent *Escherichia coli* (BL21). Single clone was cultured in LB medium containing 30 μg/ml kanamycin at 37 °C to an indicated concentration (OD value of culture was 0.8-1.0), and then IPTG (1 mM) was added and the bacteria were cultured at 18 °C for another 12 h. Cultured bacteria were collected and lysed, and the his-tag fusion proteins were adsorbed by Ni-Charged Resin (GenScript, L0066-25) and then eluted by imidazole (400 nM).

### In vitro kinase-assay

Purified recombinant proteins were firstly washed with 1× assay buffer (50 mM Tris-HCl, pH7.5, 0.1 mM EGTA, 10 mM Magnesium acetate) in ultrafiltration tubes for 5 times. The indicated proteins were mixed together to a total volume of 45 μl reaction, and placed in a 30 °C water bath for 30 s. 5 μl ATP (1 mM) was then added into the reaction, which was then vortexed and placed in a 30 °C water bath for another 20 min.

### Mass spectrometry analysis

Proteins were precipitated with 20% trichloroacetic acid (TCA) and the pellets were washed twice with 1 ml cold acetone, and dried in speed vac. Protein pellets were dissolved in 1% SDC/10 mM TCEP/40 mM CAA/Tris-HCl pH8.5. Subsequently, 1% SDC was diluted into 0.5% with water. The protein centration was measured with Pierce 660 nm protein assay reagent (Thermo, 22660). Trypsin (Sigma, T6567) was added with the ratio of 1:100 (trypsin:protein). The tubes were kept at 37 °C for 12–16 h. Peptides were desalted with SDB-RPS stage tips. For Orbitrap Fusion Lumos (Thermo), an ultra-high pressure nano-flow chromatography system (U3000, Thermo) was coupled. Liquid chromatography was performed on a reversed-phase column (40 cm × 75 μm i.d.) at 50 °C packed with Magic C18 AQ 3-μm 200- Å resin with a pulled emitter tip. A solution is 0.1% FA in $H_2O$, and B solution is 0.1% FA in ACN. In 120-min experiments, peptides were separated with a linear gradient from 0%-5% B solution within 5 min, followed by an increase to 30% B solution within 105 min and further to 35% B solution within 5 min, followed by a washing step at 95% B solution and re-equilibration. The timsTOF pro was operated in data-dependent acquisition mode. The MS raw files were subjected to Proteome Discovery software (version: 2.2.0.388) analysis. The raw files were searched against Mus Mouse database (downloaded from Uniprot). The mass tolerance of precursor and product ions were set at 10 ppm and 20 ppm with acetylation (protein N-terminal), oxidation for Methionine and phosphorylation for Serine, Threonine and Tyrosine as variable modifications. Candidate peptides of targeted proteins were systematically validated by manual inspection of spectra.

### Electron microscopy

iPCs of indicated genotypes in the iPC differentiation system at day 8 were sorted and fixed in 4% glutaraldehyde and imaged by transmission electron microscope (Hitachi, HT-7800).

### Immunization

For OVA immunization, OVA (albumin from chicken egg white, grade II, Sigma, A5253) was dissolved in PBS and 10% KAl(SO$_4$)$_2$ at 1:1 ratio, in the presence of LPS (*Escherichia coli* 055:B5,Sigma, L2880). pH was adjusted to 7.0 to form precipitate, which was utilized to immunize mice intraperitoneally (i.p.). Germinal center reaction was analyzed at day 7.5 after immunization. For NP-OVA immunization, each mouse was injected intraperitoneally (i.p.) with 50 μg NP-OVA (Biosearch Technologies, N-5051) precipitated in alum. 14 days after immunization, GCB cells and PCs (or plasmablasts) in spleen were analyzed by flow cytometry and NP-specific ASCs were measured by ELISpot assay. For memory responses, 5 μg NP-OVA precipitated in PBS was injected i.p. and the concentration of NP-specific antibodies in sera was measured by ELISA at indicated times. For NP-Ficoll immunization, each mouse was immunized with 5 μg NP-Ficoll dissolved in PBS by intraperitoneally injection, and serum NP-specific antibodies were measured by ELISA at indicated times. For NP-LPS immunization, each mouse was immunized with 50 μg NP-LPS dissolved in PBS by intraperitoneally injection, NP-specific PCs were analyzed by flow cytometry and NP-specific ASCs (IgM, IgG2b, and IgG3) were measured by ELISpot assay at indicated times.

### Immunohistochemistry

Fresh spleen from OVA/Alum/LPS-immunized mice at day 7.5 was fixed in Fixation/Permeabilization buffer (BD, 554714) overnight at 4 °C and frozen in OCT (Tissue-Tek; Sakura). 10 μm sections were cut with a micro-tome (Leica, CM1950) at −20 °C. Slides were blocked with normal rat serum and stained with AF647 anti-mouse GL7 (Biolegend; 144606), Alexa Fluor 488 anti-mouse CD3 (Biolegend; 100210), and eFluor450 anti-mouse IgD (eBioscience; 48-5993-82). Coverslips were mounted with Fluoromount-G (Southern Biotech; 0100-01) and sealed with clear nail polish. Slides were imaged on an LSM 780 C4.47 laser scanning confocal microscope (Zeiss).

### CRISPR/Cas9 screen

The candidate p38α substrates were predicted or reported by the PhosphoSitePlus® PTM website (https://www.cellsignal.com/learn-and-support/phosphositeplus-ptm-database) and previous studies[29,30]. sgRNAs targeting candidate genes were designed at the Crisprgold website (https://crisprgold.mdc-berlin.de/), synthesized, and cloned into the pMSCV-IRES-BFP vector. Retroviruses encoding sgRNAs and the BFP reporter gene were utilized to transduce cultured $Cas9^{gg/+}$or $p38α^{-/-}Cas9^{gg/+}$B cells (GFP$^+$) at day 2 in the iPC differentiation system,

**Article** https://doi.org/10.1038/s41467-022-34969-0

followed by culture for another 6 days. iPCs (CD138+) among retrovirus-transduced cells (BFP+GFP+) were analyzed by flow cytometry.

## RNA-seq and data analysis

iGCB cells at day 4 and iPCs at day 8 of indicated genotypes in the iPC differentiation system were sorted by flow cytometry. Total RNA of sorted cells was extracted with Trizol reagent (Invitrogen, 15596026) following the manufacturer's instructions. Total RNA quantity and purity were analyzed by Bioanalyzer 2100 and RNA 6000 Nano Lab Chip Kit (Agilent, 5065-4476) with RIN number >7.0. 3 μg RNA was used as input for RNA-Seq library preparations. Sequencing libraries were generated using NEB Next Ultra II RNA Library Prep Kit for Illumina (NEB, E7760) following the manufacturer's recommendations. Illumina Hi-seq system was used for sequencing of the library. Raw data (raw reads) of Fastq format were processed by Fastp software (version: 0.19.7) to remove low-quality reads. Subsequent analyses were based on high quality clean data. Reads were aligned to the reference genome using STAR (version: 020201). Cufflinks (version: 2.2.1) was used to count the read numbers mapped to each gene. FPKM (Fragments per kilobase per million) of each gene was calculated. All genes with mapped reads were used for PCA analysis[62]. Differential expression analysis of two groups was performed with a FDR value (q-value) of 0.05 and a fold change of 2 as the cut-off. Gene Ontology (GO) enrichment analysis of differentially expressed genes was implemented by the cluster Profiler R package with corrected *P*-value of 0.05 as the cut-off[53].

## Statistical data analysis

All statistical analyses were performed with GraphPad Prism (version: 8.0.2) and SPSS (version: 27.0), and *P*-values were determined by two-tailed unpaired *t*-tests in most experiments, except the *P*-values were determined by one-side hypergeometric test in Gene Ontology enrichment analysis in Fig. 4d and Supplementary Fig. 5a.

## Reporting summary

Further information on research design is available in the Nature Portfolio Reporting Summary linked to this article.

## Data availability

The RNA-seq data generated in this study have been deposited in the NCBI GEO database under accession code GSE167699. The mass spectrometry data generated in this study have been deposited in the Integrated Proteome Resources database under accession code PXD037718. Mus Mouse database used for MS analysis was downloaded from Uniprot. All plasmids and experimental materials in this study are available from the corresponding author upon request. Source data are provided with this paper.

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

## Acknowledgements
The authors thank Pengda Chen (School of life sciences, Xiamen University, China) for providing the CRISPR/Cas9-mediated sgRNA screen system and technical assistance in PCA analysis of RNA-seq data; Chenfeng Liu and Jun Xie (School of life sciences, Xiamen University, China) for technical assistance in flow cytometry; Professor Nengming Xiao for the help of plasmids and mice; Professor Kairui Mao for technical assistance in immunohistochemistry; Yaoji Liang (Biochee Biotech, Xiamen, China) for technical assistance in RNA-seq experiments; Professor Kunliang Guan for the application for *p38*⁻/⁻ 293 A cells; the Laboratory Animal Research Center of Xiamen university for in vitro fertilization service. This work was supported by the National Key R&D Program of China (2020YFA0803500 to J.H.), National Natural Science Foundation of China (81788101 and 81630042 to J.H., 31900639 to J.W. and 31770950 to W.-H.L.), the National Key R&D Program of China (2018YFA0107300 to J.W.), the 111 Project (B12001 to J.H.), Research unit fund from the Chinese Academy of Medical Sciences (2019RU054 to J.H.).

## Author contributions

J.W., K.Y., S.C., X.Z., L.H., and F.L. performed most of the experiments. J.W., K.Y., and W.-H.L. conceptualized and designed the project, interpreted the results and prepared the manuscript. S.W. generated some gene mutant mice. J.W., C.X., W.-H.L., and J.H. wrote the manuscript with contribution from all authors. J.W., C.X., W.-H.L., and J.H. contributed to data interpretation and discussions. W.-H.L., and J.H. conceived and supervised the study.

## Competing interests

The authors declare no competing interests.
