## [Peer Review File · Nature Communications]

A p38 α -BLIMP1 signalling pathway is essential for plasma cell differentiationEditorial Note: Parts of this Peer Review File have been redacted as indicated to remove third-party material where no permission to publish could be obtained.

REVIEWER COMMENTS

Reviewer #1 (Remarks to the Author):

The manuscript of Wu et al. investigates the role of p38 α in B cell development. Using mutant mice with B cell-specific deletion of p38 α , the authors describe that p38 α is not implicated in B cell development and GCB cell generation. However, p38 α seems to be implicated in plasma cell (PC) differentiation and IgG1 production. The authors find that p38 α -/- B cells exhibit a decreased BLIMP1 expression compared to WT cells. BLIMP1 is a transcription factor that regulates PC differentiation, among other functions in B and T cells. They present further data analyzing the role of p38 α substrates in PC differentiation and BLIMP1 expression, and conclude that p38 α activity controls PC differentiation by upregulating Blimp1 transcription via TCF3, TCF4, and IRF4. The work is interesting, although the role of p38 α in B cell differentiation and antibody production has been described by other authors. The main problems of this work are that many of the conclusions are based on correlation of data only and there is a lack of mechanistic insight on the role of p38 α in PC differentiation. Data presented are often convincing, but statistical significance in some experiments is questionable. In general, the "n" of some key experiments is low (n=3) to draw solid conclusions, which dampens enthusiasm for the study.

These points, as well as other comments that are listed below need to be addressed in order to make this paper suitable for publication:

1. The authors claim that p38 α activity controls PC differentiation by upregulating Blimp1 transcription via TCF3, TCF4, and IRF4. They show that knockdown of TCF3, TCF4 and IRF4 protein in WT cells decrease BLIMP1 protein levels, but they did not show if BLIMP1 levels (protein or mRNA) were restored in p38 α -/- B cells. by overexpressing TCF3, TCF4 or IRF4. Also, how does p38 α regulate TCF3, TCF4 and IRF4 activity? The authors do not show if p38 α phosphorylate any of those proteins in their experimental conditions, this should be addressed. Does p38 α regulate TCF3, TCF4 or IRF4 expression? This is not clear in this work. The signal on the blots presented in supplementary Fig S7D are saturated and it is difficult to see if the lack of p38 α is affecting the endogenous levels of TCF3, TCF4 and IRF4 in p38 α -/- B cells. Is it possible that the lack of p38 α is causing a decrease in the level of those proteins to a degree that is enough to block their action? If iPC differentiation is dependent on p38 α activity, why the overexpression of TCF3, TCF4 or IRF4 in p38 α -/- B cells increases iPC differentiation? The role of the axis p38 α -MEF2C in B cells development and IgG1 production has been described. MEF2C is a well-established p38 α substrate that is implicated in TCF3 expression and also in BLIMP1 action (at least in T cells). Why the authors have ignored MEF2C in their study? Does MEF2C phosphorylation regulate TCF3/TCF4/IRF4 activity, BLIMP1 levels and iPC differentiation in their experimental conditions?
2. On page 6 the authors mention that "p38 α fl/flCD19cre mice had significantly less production of total and high-affinity NP-specific IgG1 but not NP-specific IgM (Fig. 1E). Reduction in NP-specific IgG1 levels was also observed in secondary antibody responses of p38 α fl/flCD19cre mice (Fig. 1E)" and they claim that "These data also explain the low basal IgG1 level in p38 α fl/flCD19cre mice (Supplementary Fig. 1C)". Data immunization does not explain the results in basal conditions, they just correlate. Why do these mice have less IgG1 in basal conditions? Has this anything to do with a defect in PC or in class switch recombination? Have they looked into the IgM and IgGs in PC supernatant? Since p38 α fl/flCD19cre mouse has a defect in PC differentiation, one could think that they would have a general decrease in antibody production.
3. On page 6 the authors mention "On the other hand, p38 α fl/flCD19cre mice showed normal antibody responses after NP-Ficoll immunization (Supplementary Fig. 1D), suggesting that p38 α in B cells was not involved in type II T cell-independent immune response" However, they did not present the data of IgG1 production, so, how can they claim that T cell-independent immune response is not affected in p38 α -KO B cell? Additionally, what happens to the in vivo PC differentiation after LPS or NP-Ficoll (T-independent) immunization?
4. Could you better explain the lower panel of Fig 1D, this is very confusing. Also, does IgM ELISpot assay give similar results in p38 α -/- and WT?

5. In the in vitro differentiation system, what type of cells are the non-iPC cells? according to the principal component analysis, p38 α ^{-/-} and WT non-iPCs also exhibited very distinct gene expression profiles, this should be mentioned in the text.
6. The examination of the involvement of other p38 family members in PC generation, should be mentioned in Supplementary Figure S1. Also, a control showing the lack of expression of the different p38s should be shown, or alternatively refer to the work where these cells have been used before.
7. In general, transfection controls are missing. For example, in Figure 2E, 4, 5C... etc controls showing the p38 α , p38 α mutant, BLIMP1 expression are missing.
8. Proliferation data presented in Supplementary Fig S3B are not clear, are these cells proliferating? Controls are missing. Do you find the same results using a different method to measure cell proliferation?
9. On page 8 it would help if the authors explain, in one sentence, the rationale for generating p38^{af1/fl}CD19^{cre}Blimp1^{gfp/+} mouse. On the same page, the authors claim that "p38 α activation promotes BLIMP1 expression and plasma cell differentiation", but what they really show in Fig 3 is a correlation between the expression of p38 α and a delay in BLIMP1 expression, so, their data only suggest that p38 α expression promotes BLIMP1 expression.
10. It would be nice if the authors could explain in the text something about the proteins analyzed in Fig 5E.
11. I think some results are misplaced, results in Fig 5A, 5B and Supplementary Figure 5 should be moved to the section "p38 α regulates BLIMP1 expression and plasma cell differentiation".
12. Is the effect on ER-related transcriptome a consequence of the lower levels of BLIMP1 and decreased differentiation? This should be clarified and discussed in the manuscript.
13. Supplementary Figure 5C: Needs quantification to see if protein stability changes or is the same in p38 α ^{-/-} and WT cells.
14. Could you include the English version of the following webpage?
<https://www.cellsignal.cn/learn-and-support/phosphositeplus-ptm-database>
15. Lack of references across the discussion. Also, in the discussion, there are several points: (1) Lines 375-377: could the authors explain which other approaches were used in other studies? (2) Lines 385-386: could the authors explain what they mean in the following sentence? "Thus, the underlying mechanism of aberrant B cells development and GCB cells generation by the treatment of p38 inhibitors needs further investigations" (3) Lines 393-394, in the sentence "Since the germinal center reaction was not well-studied in the study" do the authors mean extensively-studied instead? Please explain (1) and (2) and correct (3).
16. Many references are misquoted, these are just two examples: (1) On pages 4 and 5, the authors mentioned that "Previous studies suggested p38 α and p38 β were highly expressed in follicular B cells and p38 α was the most abundant one in PCs" according to references 29 (Khiem et al 2008 PNAS) and 30 (Craxton et al., 1998 J. Immunol); however, in those papers the p38 α and p38 β expression was not described. (2) On methods they claim that p38^{af1/fl}, p38 ^{β fl/fl}, p38 ^{β -/-}, p38 ^{γ -/-}, p38 ^{δ -/-} mice were previously described in 50 (Wu et al 2013 Cell Res) and 51 (Zheng et al 2011 Nat Cell Biol), this is wrong. Also, the reference Streicher et al., 2014 is missing in the list of references. A general revision of the references is needed in this manuscript.
17. Supplementary Fig S1, explain what is the Pro-pre B cell population, did you mean Pro and Pre cell populations? More data of the gating used in the analysis should be provided.
18. On page 4 line 68 the words "has been used for" are repeated

Reviewer #2 (Remarks to the Author):

In their study entitled "A p38 α -BLIMP1 signaling pathway is essential for plasma cell differentiation" Wu et al convincingly demonstrate a key role for p38 α in plasmacytic differentiation. The results are novel and highly relevant, as p38 α inhibition has therapeutic value in B cell-driven autoimmunity. The experiments are of very high quality and are extremely comprehensive. The authors use relevant in vivo models and take full advantage of a highly suitable in vitro model of plasmacytic differentiation for detailed mechanistic studies. Wu and colleagues generated mice lacking p38 α specifically in B cells or in the entire hematopoietic system. They carefully analyzed B cell-dependent immune responses in these mice and found a fundamental defect in plasma cell differentiation and T cell dependent antibody responses, although type II independent antibody responses remained intact. They then employed the 40LB cell culture system to study plasmacytic differentiation in detail. They demonstrated that p38 is the only family member whose sole deficiency deregulates plasmacytic differentiation and that its kinase activity is required. Wu et al show that p38 α regulates Blimp1 transcriptionally and performed a CRISPR screen of p38 α targets to identify the relevant downstream mediators. In a series of experiments involving expression of both splice isoforms and knockout they validate Tcf3 as important downstream mediator of p38 α signaling in plasmacytic differentiation. Finally, Tcf4 and Irf4 are further validated in their role regulating plasmacytic differentiation and Wu et al present evidence for a Tcf3/Tcf4/IRF4-dependent induction of Blimp1 expression downstream of p38 α .

I have only very limited questions and comments as in my opinion this important study can essentially stand as it is.

Major comments

1) What is the role of p38 α mediated phosphorylation of Tcf3? p38 α loss does not affect protein levels of Tcf3, Tcf4 and Irf4 only mildly. But overexpression of these proteins compensates for absence of p38 α in plasmacytic differentiation. Furthermore, the Tcf3-S140A mutant rescues plasma cell differentiation similarly to wild-type Tcf3. Therefore, the role of p38 α -mediated phosphorylation remains unclear. These aspects are clearly discussed in the manuscript and I do not think the authors have to resolve these questions in the present manuscript. But they should explore whether massive overexpression occurs in their rescue experiments and whether this could explain a (partial) independence of phosphorylation (of Tcf3 or a putative upstream regulator) in this setting. The degree of overexpression of the factors (p38, Tcf3, Tcf4, Irf4) in the rescue experiments should be shown compared to the endogenous levels.

2) GCB cell production seems dramatically enhanced in mice lacking both p38 α and p38 β in B cells (not slightly as mentioned in the discussion). This comes up only in the discussion and plasma cells are not shown in supplemental figure 8. The authors should either show plasma cells here, as surely they looked at it, or omit this figure as it exceeds the scope of the present manuscript to fully analyze p38 α /beta double deficiency in B cells.

Minor comments

3) In figure S1 splenic B1 cells should be called B1a, as they are gated on expression of CD5. Also in S8B.

4) In figure 1D the same data are shown twice with a different scale? Once is sufficient in my opinion.

5) In figure 1E in the secondary immune response the overall IgG1 response (NP29) is lower than the high affinity response (NP7)?

6) In figure S2 I would remove the statistical analyses from experiment with n = 2. Also applies to 5J.

S2D: how long were these B cells cultured, please indicate in the figure legend.

7) Figure 5H: can the luciferase reporter be shown in the supplemental material?

8) Figure 7B: when were the cells retrovirally transduced to express p38alpha and its mutants? Same question for 7C.

9) Is the full gating strategy to identify B cell subsets always indicated in the figure legends? For example, GCB cells in Figure 1C were only identified by NP, Fas and GL-7?

10) Reference 31 indeed shows a reduction in IgG1 levels in p38 deficiency, congruent with the present study. This should be reflected in the discussion.

Reviewer #3 (Remarks to the Author):

The present manuscript by Wu and Han and their colleagues seeks to understand the role of p38alpha (p38a, MAPK14) in plasma cell (PC) development, as the literature on a possible role of this kinase in this process is unknown. The authors addressed this question by employing a conditional Mapk14 allele encoding p38a and deleting it in B cells (CD19-Cre mice). The main finding that, although germinal center (GC) B cell development is not affected, PC development is markedly impaired, led the authors to elucidate the underlying molecular mechanism in an established in vitro assay that mimics the GC to PC development – both iGC and GC B cells are being cultured and experimentally manipulated. An p38a-BLIMP1 axis with several intermediaries is identified that is critical for the transmission of developmental signals in the GC response and upon LPS stimulation (T-independent type I), but not in the T-independent type II response, demonstrating that not all PC development depends on p38a.

Overall, from Fig. 2 onwards, this manuscript contains high quality data that convincingly uncover a role for p38a in the BLIMP1-dependent development of PCs. This is important information for the field of adaptive immunity in general and PC biology in particular, and the results may be exploited and further developed into potential treatments of PC-associated diseases, including certain autoimmune conditions. However, as detailed below, there are a number of unclear or potentially contentious issues that would require attention.

1) On the assumption that in Fig. 1A top (the GC plots), it is gated on B cells, although this is not indicated in the figure legend, the percentage of GC B cells is unusually high, appr. 3 to 4 times higher than expected. Is this because of the unusual OVA/Alumn/LPS conjugate used for the immunization? Why was this conjugate used, whereas in later experiments, only OVA/Alumn was used, which is the normal way (excluding LPS).

2) On the assumption that in Fig. 1A bottom (the PC plots), it is gated on splenic mononuclear cells, although this is not indicated in the figure legend, the percentage of CD138+ plasma cells is unusually high, appr. 5 to 10 times higher than expected. However, CD138+ plasma cells should express 10-fold higher cell surface amounts of CD138 than in the population shown (one log higher in the FACS plot). Since there is no such population: were the plots gated on B cells, so that PC would be gated out? The bottom line is that as shown, it is not convincing that PC percentages are measured, because there are too many and not at the usual CD138 level. Or does the LPS lead to this phenotype? Unlikely. Some response to these points appears necessary.

3) Fig. 1C: What exactly is shown in the plots, what is the gating strategy? This should be indicated in the figure legends and on top of the plots.

4) Fig. 1C: CD138+ PCs do not express surface Ig, so what is measured is probably CD138+ plasmablast. So the plots in Fig. 1C bottom are not informative, in contrast to the ELISPOT shown in Fig. 1D which provide convincing data that PC development is affected.

5) Supplementary Fig. 1C: that the basal serum level of IgG1 was decreased in the p38a CD19-conditional mice is in disagreement with the data from the blastocyst complementation study which showed no difference in IgG1 levels in unimmunized mice (ref. 31). However, the control mice have a very large range of serum IgG1 concentrations, and it seems there are actually two subpopulations, with the high concentration ones perhaps representing outliers, letting one wonder if the latter mice were perhaps infected with something? It does not appear that from these results it can be concluded that there is less IgG1 baseline in the conditional p38a mice (see also #8).

6) Fig. 2B: Information on the identity of the differentially expressed genes should be provided in a Supplementary Table. How do we know the analysis was performed correctly? Knowing which

genes are differentially expressed may provide answer to this.

7) Discussion line 381: "Gene expression profiles of p38^{-/-} and WT iGCB cells are almost identical, with only 64 differentially expressed genes (Fig. 2B)." Well, 64 genes are quite a lot. Again, here it is important to know the identity of the genes and the fold-differences.

8) Discussion line 400: Again, the comparison of the serum IgG titers (baseline, without immunization) between ref. 31 and Suppl. Fig. 1C is rendered ambiguous by the apparent two groups in the p38^{fl/fl} (no Cre) control. This could be clarified by repeating the experiment shown in the figure, although overall the data on a role of p38a in the generation of GC derived PCs are so convincing that demonstrating that it does not have effect -- or a small one -- on baseline IgG1, which may well come from different sources of p38a-independent PCs, is in the view of this reviewer not essential.

9) Introduction line 50, ref. 10: these experiments were done in B-cell lymphoma lines and not mature B cells. This would need to be clarified.

10) Introduction line 91: Streicher et al. is in the wrong reference format. Perhaps the sentence should be worked over, because to say that "...dysregulated PC differentiation often leads to ... multiple myeloma" is not quite accurate, as myeloma is a tumor of fully differentiated PCs.

Point-by-point response

REVIEWER COMMENTS

Reviewer #1 (Remarks to the Author):

The manuscript of Wu et al. investigates the role of p38 α in B cell development. Using mutant mice with B cell-specific deletion of p38 α , the authors describe that p38 α is not implicated in B cell development and GCB cell generation. However, p38 α seems to be implicated in plasma cell (PC) differentiation and IgG1 production. The authors find that p38 α ^{-/-} B cells exhibit a decreased BLIMP1 expression compared to WT cells. BLIMP1 is a transcription factor that regulates PC differentiation, among other functions in B and T cells. They present further data analyzing the role of p38 α substrates in PC differentiation and BLIMP1 expression, and conclude that p38 α activity controls PC differentiation by upregulating Blimp1 transcription via TCF3, TCF4, and IRF4. The work is interesting, although the role of p38 α in B cell differentiation and antibody production has been described by other authors. The main problems of this work are that many of the conclusions are based on correlation of data only and there is a lack of mechanistic insight on the role of p38 α in PC differentiation. Data presented are often convincing, but statistical significance in some experiments is questionable. In general, the “n” of some key experiments is low (n=3) to draw solid conclusions, which dampens enthusiasm for the study.

These points, as well as other comments that are listed below need to be addressed in order to make this paper suitable for publication:

Response: We thank this reviewer very much for the very constructive comments, which helped us in improving our manuscript. We have increased n to ≥ 5 in most experiments and addressed other criticisms in the revision. The details are described below.

1. *The authors claim that p38 α activity controls PC differentiation by upregulating Blimp1 transcription via TCF3, TCF4, and IRF4. They show that knockdown of TCF3, TCF4 and IRF4 protein in WT cells decrease BLIMP1 protein levels, but they did not show if BLIMP1 levels (protein or mRNA) were restored in p38 α ^{-/-} B cells by overexpressing TCF3, TCF4 or IRF4.*

Response: The reviewer is right and we have now included data showing the protein level of BLIMP1 was partially restored in cultured p38 α ^{-/-} B cells by over-expressing TCF3, TCF4 or IRF4 in iPC differentiation system (Reviewer 1 Fig. 1, 2, 3, that are included as Fig. 5e, 6c, 6f in the revised manuscript).

Reviewer 1 Fig. 1: Immunoblot analysis of BLIMP1, IgG H-chain, Actin and p38 α in TCF3 (E47 or E12)-overexpressing cells (retrovirally transduced, GFP⁺) at day 8 in iPC culture.

Reviewer 1 Fig. 2: Immunoblot analysis of BLIMP1, IgG H-chain, Actin and p38 α in TCF4-overexpressing cells (retrovirally transduced, GFP⁺) at day 8 in iPC culture.

Reviewer 1 Fig. 3: Immunoblot analysis of BLIMP1, IgG H-chain, Actin and p38 α in IRF4-overexpressing cells (retrovirally transduced, GFP⁺) at day 8 in iPC culture.

Also, how does p38 α regulate TCF3, TCF4 and IRF4 activity? The authors do not show if p38 α phosphorylate any of those proteins in their experimental conditions, this should be addressed.

Response: Yes, we agree. We have worked very hard in addressing this question. In the revised manuscript, we show that p38 α can directly phosphorylate IRF4 and identified T267, T268, T269 and S270 residues on IRF4 as a cluster of sites that can be directly phosphorylated by p38 α , and deficiency in the phosphorylation of T267, T268, or T269 but not S270 impaired IRF4 activation (Fig. 7b, Supplementary Fig. 9d in the revised manuscript). Unfortunately, we were able to show that TCF3 and TCF4 could be directly phosphorylated on certain sites by p38 α *in vitro* but did not find the phosphorylation of any or a cluster of these sites were critical for p38 α -dependent TCF3 or TCF4 activation. We have to admit that we did not solve the question of whether p38 α directly regulates the

activities of TCF3 and TCF4. Though my laboratory has pretty long-time experience and sufficient experimental tools in mapping phosphorylation sites, I cannot foresee the day that we can thoroughly resolve this question that how p38 α upregulated the activity of TCF3 and TCF4 during PC differentiation. We wish the reviewer and editor would agree that we shall leave this question for future investigation. Below is a brief description of what we have done.

--To examine whether TCF3, TCF4, or IRF4 can be directly phosphorylated by p38 α , we prepared recombinant TCF3, TCF4 and IRF4 proteins as substrates using *E.coli* expression system and performed *in vitro* kinase assay using recombinant p38 α and MKK6E as kinases. The phosphorylation sites of p38 α were determined by MS and shown in Reviewer 1 Fig. 7a, 8a, 9a. p38 α can phosphorylate these three proteins efficiently *in vitro* and a number of phosphorylation sites were identified.

--To further identify the functional phosphorylation sites, we constructed and evaluated luciferase reporters. When luciferase reporters driven by the TCF3, TCF4, or IRF4 binding sites respectively from the *Blimp1* enhance region were introduced into the iPC differentiation system, we observed that p38 α but not its kinase-dead mutants (K53M or T180A/Y182F) were able to reconstitute endogenous p38 α in mediating the upregulation of those luciferase reporters (Reviewer 1 Fig. 4, Fig. 7a in the revised manuscript), indicating that reporters can be used in our studies. We also tested reporters used by others in the field, that are a reporter commonly used for TCF3 and TCF4 dependent transcription (PMID:1899229) and one for IRF4 dependent gene expression (PMID:19433799, 23804646) (named E-box reporter and IRF4 reporter respectively), and obtained similar results (Reviewer 1 Fig. 5, Supplementary Fig. 9b in the revised manuscript). p38 α kinase activity-dependent enhancement of TCF3/TCF4/IRF4 transcription activity to these reporters were observed in p38 α , β , γ and δ null 293A (p38 α ^{-/-} 293A) cells (Reviewer 1 Fig. 6, Supplementary Fig. 9c in the revised manuscript).

--To investigate whether the phosphorylation of TCF3 by p38 α activates TCF3, we tested whether co-expression of p38 α or p38 α +MKK6E with TCF3 (either E47 or E12 isoform) in p38 α ^{-/-} 293A cells can increase TCF3 reporter activity. We detected p38 α dependent enhancement of reporter expression (Reviewer 1 Fig. 6). However, the mutation of each one of the potential p38 α phosphorylation sites (Reviewer 1 Fig. 7b), as well as each of the S/T followed by Pro (conserved MAP kinase phosphorylation sites, total 13 in E47) (Reviewer 1 Fig. 7c) did not affect p38 α -mediated enhancement of TCF3 (E47) dependent transcription. Several other cluster mutations were also tried but no effect was found (Reviewer 1 Fig. 7d).

--The same approach was used in addressing the mechanism of how p38 α regulates TCF4 activity and also did not have any progress (Reviewer 1 Fig. 8).

AT THIS END, we shall state that we still do not know how p38 α regulates TCF3/TCF4. We discussed this issue in detail in the revised manuscript.

--As for IRF4, we found T267A, T268A and T269A mutants of IRF4 were much weaker than wildtype IRF4 in response to p38 α in driving IRF4 dependent reporter gene expression. The triple T to A mutant (T267A/T268A/T269A) of IRF4 completely lost the transcription activity, suggesting these three sites are functional phosphorylation sites by

p38 α (Reviewer 1 Fig. 9, Fig. 7b in the revised manuscript). S270 next to these three T residues is most likely to be another phosphorylation site in cells as amino acid 271 is a Pro but its A mutation had no effect on p38 α mediated enhancement of IRF4 transcription activity. The role of IRF4 phosphorylation on T267, T268 and T269 was further demonstrated by the data that these single T to A mutants of IRF4 failed to restore iPC generation and BLIMP1 expression in cultured *Irf4*^{-/-} B cells (Reviewer 1 Fig. 10, Fig. 7c in the revised manuscript). Thus, our results demonstrated that IRF4 phosphorylation at T267, T268, T269 by p38 α is the activation mechanism of IRF4 in iPC differentiation.

Luciferase reporter of binding site in *Prdm1* locus

Reviewer 1 Fig. 4: Retroviruses encoding p38 α or p38 α kinase-dead mutants (K53M or T180A/Y182F), and indicated TCF3, TCF4 or IRF4 luciferase reporter were co-transduced into cultured p38 α ^{-/-} B cells at day 2 and luciferase activity in retrovirus-infected cells were analyzed at day 8 in iPC differentiation system.

Reviewer 1 Fig. 5: p38 α KO and WT B cells were transduced with indicated luciferase reporter and cultured in the iPC differentiation system, and the luciferase activity in transduced cell were analyzed at indicated time. Samples were triplicated.

p38^{-/-} cells (293A)

Reviewer 1 Fig. 6: p38^{-/-} 293A cells were transfected with indicated plasmids and the luciferase activities of TCF3(E47 or E12)/TCF4/IRF4 in transfected cells were examined 24 hours after transfection.

a

TCF3 (E47)

P : phosphorylation residues identified by MS analysis
 — Serine or Threonine residues that precede a Proline residue

b

c

d

Reviewer 1 Fig. 7: (a) Schematic outline of the p38α phosphorylated residues on E47 identified by MS (red) and converted MAP kinase phosphorylation sites on E47 (black). **(b to d)** p38^{-/-} 293A cells were transfected with indicated plasmids including E-box promoter and the luciferase activity of E47 mutants expressing cells was examined 24 hours after transfection.

a

TCF4

P: phosphorylation residues identified by MS analysis
 — Serine or Threonine residues that precede a Proline residue

b

Reviewer 1 Fig. 8: (a) Schematic outline of the p38α phosphorylated residues on TCF4 identified by MS (red) and converted MAP kinase phosphorylation sites on TCF4 (black). **(b)** p38^{-/-} 293A cells were transfected with indicated plasmids including E-box promoter and the luciferase activity of TCF4 mutants expressing cells was examined 24 hours after transfection.

a

P : phosphorylation residues identified by MS analysis

b

Reviewer 1 Fig. 9: (a) Schematic outline of the p38α phosphorylated residues on IRF4 identified by MS (red). **(b)** *p38*^{-/-} 293A cells were transfected with indicated plasmids including IRF4 reporter and the luciferase activity of IRF4 and IRF4 mutants expressing cells was examined 24 hours after transfection, the protein levels of p-p38α, p38α, IRF4 and Actin in these cells were examined at 24 hours after transfection.

Reviewer 1 Fig. 10: Retroviruses encoding IRF4 mutants (T267A, T268A, T269A) were

transduced into cultured *Irf4*^{-/-} B cells at day 2, iPCs in retrovirally transduced cells (CD19⁺GFP⁺CD138⁺) 6 days after transduction were analyzed by flow cytometry, protein levels of IRF4, IgG H-chain, BLIMP1 and Actin were analyzed by Western blot.

Does p38α regulate TCF3, TCF4 or IRF4 expression? This is not clear in this work. The signal on the blots presented in supplementary Fig S7D are saturated and it is difficult to see if the lack of p38α is affecting the endogenous levels of TCF3, TCF4 and IRF4 in p38α^{-/-} B cells. Is it possible that the lack of p38α is causing a decrease in the level of those proteins to a degree that is enough to block their action?

Response: We apologize for the confusion we made. We have replaced Fig. S7D with another experimental data and semi-quantitated these proteins by measuring densities of bands in immunoblots (Reviewer 1 Fig. 11, Supplementary Fig. 7c and 7d in the revised manuscript). *p38α* deficiency had no significant effects on TCF3 and TCF4 protein levels, but IRF4 protein level was decreased in cultured *p38α*^{-/-} B cells. Previous studies suggested that BLIMP1 and IRF4 formed a positive regulatory circuit during plasma cell differentiation (PMID: 23684984, 26779602). It is hence conceivable that the decrease in IRF4 expression levels in cultured *p38α*^{-/-} B cells resulted from decreased BLIMP1 expression. Indeed, we observed decreased IRF4 protein level in cultured *Blimp1*^{-/-} B cells (Reviewer 1 Fig. 12, Supplementary Fig. 7e in the revised manuscript), which is consistent with previous publications showing that BLIMP1 and IRF4 positively regulate each other during plasma cell differentiation. We have included these results in the revised manuscript.

Reviewer 1 Fig. 11: Immunoblot analysis of TCF3, TCF4, IRF4, BLIMP1, IgG H-chain and

Actin in *p38α* KO and WT iGCB cells transferred onto IL-21 feeder cells and cultured for indicated amounts of time. The densities of the protein bands are presented.

Reviewer 1 Fig. 12: Immunoblot analysis of IgG H-chain, IRF4, BLIMP1, and Actin in cultured *Blimp1* WT and KO B cells at day 8 in iPC culture. Flow cytometry analysis of the percentage of iPCs (CD19⁺CD138⁺) at day 8 of culture.

If iPC differentiation is dependent on p38α activity, why the overexpression of TCF3, TCF4 or IRF4 in p38α^{-/-} B cells increases iPC differentiation?

Response: The reviewer raised a critical question and we apologize for having not explained this important issue thoroughly. The promoting effect on plasma cell differentiation of TCF3, TCF4 or IRF4 overexpression in cultured *p38α^{-/-}* B cells most likely resulted from the high-level basal activity of these over-expressed transcription factors. To support this interpretation, we tested the dose effect of these transcription factors in *p38^{-/-}* 293A cells. Indeed, the transcriptional activity of TCF3, TCF4, and IRF4 was dose-dependently increased based on their corresponding reporter assays (Reviewer 1 Fig. 13). We also showed that the transcriptional activity of these transcription factors can be increased by *p38α* (Reviewer 1 Fig. 14). Thus, overexpression of TCF3, TCF4 or IRF4 in cultured *p38α^{-/-}* B cells most likely partially mimicked the activation of these transcription factors by *p38α* during plasma cell differentiation.

Reviewer 1 Fig. 13: $p38^{-/-}$ 293A cells were transfected with indicated plasmids, the luciferase activity and protein levels of TCF3/TCF4/IRF4 in transfected cells were examined 24 hours after transfection.

Reviewer 1 Fig. 14: $p38^{-/-}$ 293A cells were transfected with indicated plasmids and the luciferase activity of transfected cells was examined at 24 hours after transfection.

The role of the axis p38 α -MEF2C in B cells development and IgG1 production has been described. MEF2C is a well-established p38 α substrate that is implicated in TCF3 expression and also in BLIMP1 action (at least in T cells) Why the authors have ignored MEF2C in their study? Does MEF2C phosphorylation regulate TCF3/TCF4/IRF4 activity, BLIMP1 levels and iPC differentiation in their experimental conditions?

Response: This is an excellent question. We had included the results of *Mef2c* in the sgRNA screen of potential p38 α substrates for their role in iPC differentiation (Fig. 5E in our original manuscript, which was changed to Fig. 5b in the revised manuscript). *Mef2c* was ranked third in reducing iPC generation in this screen, but the effect was significantly less than that of *Tcf3* knockout in this screen. Since knockout of *Mef2c* reduces the viability of B cells in this *in vitro* differentiation system (see below for more discussion on this issue), the criteria used in this screen may not be favorable for MEF2C. We nonetheless have carefully addressed the role of MEF2C in p38 α -promoted PC differentiation, and our data suggested that MEF2C did not function downstream of p38 α to regulate TCF3/TCF4/IRF4 activity, BLIMP1 expression and plasma cell differentiation.

Using an *in vitro* system, Kong et al showed that in cooperating with EBF1-MEF2C plays an important role in B cell generation and lymphoid versus myeloid fate decision and a p38 inhibitor can block this process (PMID: 33184436). However, B cell-specific *Mef2c* knockout mice (*Mef2c^{fl/fl}CD19^{cre}*) generated by Khiem et al (PMID:18955699) and Wilke et al (PMID:18438409) exhibit normal B cell development in bone marrow and spleen. They also showed that T-cell dependent antibody response in *Mef2c^{fl/fl}CD19^{cre}* mice was defective, as evidenced by decreased serum IgG1 titers and impaired germinal center

formation. They also demonstrated that MEF2C was critical for BCR-stimulated naïve B cell proliferation and survival, but not for proliferation and survival induced by other stimuli. It needs to mention that neither study showed data on the involvement of p38 α in B cell generation or IgG1 production, and that Khiem et al only showed that BCR-stimulated MEF2C activity could be blocked by p38 inhibitors *in vitro*.

The role of p38 α in B cell generation and ligand-stimulated naïve B proliferation has been studied by Kim et al (PMID:15661878). By generating p38 $\alpha^{-/-}$ chimeric mice through ES-cell mediated RAG-deficient blastocyst complementation technology, they found that p38 α is dispensable for lymphocyte development and proliferation. Consistent with the results of Kim et al, we found that B cell and HSC-specific deletion of p38 α (p38 $\alpha^{fl/fl}CD19^{cre}$ and p38 $\alpha^{fl/fl}Vav^{cre}$ mice, respectively) did not affect the amount of B cells, ligand-stimulated naïve B proliferation and germinal center B cell formation in mice (Fig. 1a-1c, 2a, S1a, S1b, S3b, S3c in the original manuscript, that are Fig. 1a-1c, 2a, Supplementary Fig. 1a, 1b, 4a, 4b in the revised manuscript), suggesting that p38 α does not play indispensable roles in these processes. The normal GCB cells generation in p38 $\alpha^{fl/fl}CD19^{cre}$ mice is different from the mice with genetic deletion of *Mef2c* in B cells.

Different from *Mef2c* deficiency, B cell specific p38 α knockout in mice did not influence GCB cell generation (Fig. 1 in the original manuscript, which is Fig. 1 in the revised manuscript). Knock out p38 α by CRISPR/Cas9 in cultured B cells had no effect on B cell number, whereas knocking out *Mef2c* in cultured B cells significantly decreased cell number (Reviewer 1 Fig. 15, Supplementary Fig. 8a in the revised manuscript), most likely resulting from cell death similar to previously reported data in *Mef2c^{fl/fl}CD19^{cre}* mice (PMID:18438409, 18955699). These data again indicated that the roles of p38 α and MEF2C in GCB cells generation and PC differentiation have differences.

It was reported that MEF2C is required to sustain the activation of Tregs and the transcriptional program activated by BLIMP1 (PMID:34290714). However, ectopic expression MEF2C, MEF2C-3E(T293E/T300E/S387E) or MEF2C-3D in cultured p38 $\alpha^{-/-}$ B cells had no effect on plasma cell differentiation (Reviewer 1 Fig. 16, which is Supplementary Fig. 8b in the revised manuscript). Moreover, CRISPR/Cas9-mediated knock-out of *Mef2c* in p38 $\alpha^{-/-}$ and WT iGCB cells decreased plasma cell differentiation in the iPC differentiation system, suggesting additive effects of MEF2C and p38 α in this process (Reviewer 1 Fig. 17, which is Supplementary Fig. 8c in the revised manuscript). Collectively, these data suggest that at least part of p38 α ' and MEF2C's functions are not in the same signaling axis during GCB cell generation and PC differentiation.

Reviewer 1 Fig. 15: Retroviruses encoding NTC sgRNA or sgRNA targeting to *p38α* or *Mef2c* (BFP⁺) were transduced into cultured B cells (Cas9-expressing B cells (GFP⁺) mixed with WT B cells (GFP⁻) at 1:1 ratio) at day 2 in iPC differentiation system. The percentages of cultured BFP⁺GFP⁻ B cells and BFP⁺GFP⁺ B cells of indicated genotypes were analyzed at day 8 by flow cytometry, and the ratio of *p38α* or *Mef2c* sgRNA transduced BFP⁺GFP⁺ B cells to NTC transduced BFP⁺GFP⁺ B cells was shown. NTC, non-targeting control.

Reviewer 1 Fig. 16. Cultured *p38α*^{-/-} B cells were transduced with retroviruses encoding MEF2C or related mutants at day 2. iPCs among retrovirus-transduced B cells (CD19⁺GFP⁺CD138⁺) were analyzed by flow cytometry, and MEF2C and Actin in retrovirus-transduced B cells (CD19⁺GFP⁺) were analyzed by immunoblot at day 8 in iPC culture.

Reviewer 1 Fig. 17: *p38α*^{fl/fl}*Vav*^{cre}*Cas9*^{tg/+} and *p38α*^{fl/fl}*Cas9*^{tg/+} iGCB cells were transferred onto IL-21 feeder cells and transduced with retroviruses encoding sgRNA targeting *Mef2c* 1 day later. The percentage of iPCs in retrovirus-transduced B cells (CD19⁺GFP⁺BFP⁺CD138⁺) was analyzed by flow cytometry, and MEF2C and Actin in retrovirus-transduced B cells were analyzed by immunoblot 3 days after transduction. NTC, non-targeting control.

2. On page 6 the authors mention that “*p38 α /fl/CD19 cre mice had significantly less production of total and high-affinity NP-specific IgG1 but not NP-specific IgM (Fig. 1E). Reduction in NP-specific IgG1 levels was also observed in secondary antibody responses of *p38 α /fl/CD19 cre mice (Fig. 1E)” and they claim that “These data also explain the low basal IgG1 level in *p38 α /fl/CD19 cre mice (Supplementary Fig. 1C)”. Data immunization does not explain the results in basal conditions, they just correlate. Why do these mice have less IgG1 in basal conditions? Has this anything to do with a defect in PC or in class switch recombination? Have they looked into the IgM and IgGs in PC supernatant? Since *p38 α /fl/CD19 cre mouse has a defect in PC differentiation, one could think that they would have a general decrease in antibody production.****

Response: We agree with the reviewer that the NP-OVA/Alum-immunization results of *p38 α ^{fl/fl}CD19^{cre}* mice in Fig. 1e and the basal IgG1 level in *p38 α ^{fl/fl}CD19^{cre}* mice in Fig. S1c in original manuscript were the only correlation. We have removed related statement in the revised manuscript.

The reviewer raised an excellent question which led us to think carefully about basal IgG1 level. The basal IgG1 expression in our experimental mice should be primarily induced by microorganisms lived with the mice as it is known that the basal IgG1 level was significant lower in germ-free mice (PMID:2606142). One would speculate that the PC generation in non-immunized mice or immunized mice may share similar mechanisms, and as the reviewer pointed out that *p38 α ^{fl/fl}CD19^{cre}* mouse would have a general decrease in antibody production. Because of limited germinal center reaction and PC numbers *in vivo*, it is hard to study class-switch recombination (CSR) in non-immunized mice. Nevertheless, we could show that the CSR in *p38 α ^{-/-}* and WT iGCBs at day 4 in iPC differentiation system was similar (Reviewer 1 Fig. 18). Since we also observed a decrease in the number of PCs in non-immunized *p38 α ^{fl/fl}CD19^{cre}* mice (Reviewer 1 Fig. 19, which is Supplementary Fig. 1d in the revised manuscript), the decreased basal IgG1 in *p38 α ^{fl/fl}CD19^{cre}* mice could due to at least in part the defect in PC generation.

We noticed that only basal IgG1 level had statistically significant difference between *p38 α ^{fl/fl}* and *p38 α ^{fl/fl}CD19^{cre}* mice, while the basal levels of other IgG isotypes (IgG2a, IgG2b and IgG3) had no statistically significant difference. IgM level also had no statistical relevant difference between *p38 α ^{fl/fl}* and *p38 α ^{fl/fl}CD19^{cre}* mice (Reviewer 1 Fig. 20, Supplementary Fig. 1c in the revised manuscript). We followed the reviewer’s suggestion to measure IgGs, IgM and IgE concentrations in supernatants of iPC culture and found that IgG1 and IgE levels were significantly lower in the culture of *p38 α ^{-/-}* B cells in comparison with their WT counterparts (Reviewer 1 Fig. 21, Fig. 2a in the revised manuscript). IgG2, IgG3, and IgM was not detected in our iPC differentiation system, which is expected since it was known that most cultured B cells differentiated into IgG1- or IgE- producing iPCs in the iPC differentiation system (PMID: 21897376). We mentioned this when the iPC differentiation system was introduced in the revised manuscript (lines: 150-151). The reviewer is right that we shall not make conclusion by speculation. We have added the new data related to basal antibodies into the revised manuscript.

Reviewer 1 Fig. 18: $p38\alpha^{-/-}$ and WT B cells were cultured on IL-4 feeder cells, and CSR in iGCB cells were analyzed by flow cytometry at day 4.

Reviewer 1 Fig. 19: Flow cytometry analysis of GCB cells and PCs in the spleen of non-immunized $p38\alpha^{fl/fl}CD19^{cre}$ and $p38\alpha^{fl/fl}$ mice (8-10 weeks, n=7 per group). Summary of the percentage and number of GCB cells and PCs.

Reviewer 1 Fig. 20: Serum immunoglobulin (Ig) levels in non-immunized $p38\alpha^{fl/fl}CD19^{cre}$ and $p38\alpha^{fl/fl}$ mice (8-10 weeks, n=8 per group) were determined by ELISA.

Reviewer 1 Fig. 21: Supernatant IgG1 and IgE of $p38\alpha$ KO and WT B cells at day 8 of iPC culture were measured by ELISA.

3. On page 6 the authors mention "On the other hand, $p38\alpha^{fl/fl}CD19^{cre}$ mice showed normal antibody responses after NP-Ficoll immunization (Supplementary Fig. 1D), suggesting that $p38\alpha$ in B cells was not involved in type II T cell-independent immune

response” However, they did not present the data of IgG1 production, so, how can they claim that T cell-independent immune response is not affected in p38α-KO B cell? Additionally, what happens to the *in vivo* PC differentiation after LPS or NP-Ficoll (T-independent) immunization?

Response: We are sorry that we did not describe these issues clearly. Previous studies have shown that IgG3 was the dominant isoform of antibody in NP-Ficoll-immunized mice, while NP-specific antibodies of other isoforms (IgG1, IgG2a, IgG2b) were undetectable (Reviewer 1 Fig. 22, PMID:28771702). Consistently, we found that serum IgG1 levels were very low and about the same in NP-Ficoll-immunized *p38α^{fl/fl}CD19^{cre}* and *p38α^{fl/fl}* mice (Reviewer 1 Fig. 23). We did detect significant amounts of anti-NP IgG3 but no difference in the levels of the IgG3 between *p38α^{fl/fl}CD19^{cre}* and *p38α^{fl/fl}* mice can be found (Supplementary Fig. 1d in original manuscript, which is Supplementary Fig. 2b in the revised manuscript). These data suggested that *p38α* deficiency does not affect type II T cell-independent immune response.

We also analyzed the *in vivo* PC generation after NP-LPS or NP-Ficoll immunization as suggested by the reviewer, and found that PC generation was significantly decreased in *p38α^{fl/fl}CD19^{cre}* mice after NP-LPS immunization (Reviewer 1 Fig. 24, which is Supplementary Fig. 2a in the revised manuscript), but was comparable in NP-Ficoll-immunized mice (Reviewer 1 Fig. 25, which is Supplementary Fig. 2c in the revised manuscript). These results were included in the revised manuscript.

[REDACTED]

Reviewer 1 Fig. 22: (Which is Fig. 3E in PMID:28771702): NP-specific IgG responses were measured at 5, 12, and 26 days after NP-Ficoll immunization in C57BL/6 mice.

Reviewer 1 Fig. 23: ELISA analysis of serum NP-specific IgG1 concentration in *p38α^{fl/fl}CD19^{cre}* and *p38α^{fl/fl}* mice immunized with NP-Ficoll (i.p) at 28 days after immunization (n=10 per group). ELISA OD values (right).

Reviewer 1 Fig. 24: Flow cytometry analysis of NP-specific PBs ($B220^{low}CD138^{+}NP^{+}$) in the spleen of $p38\alpha^{fl/fl}CD19^{cre}$ and $p38\alpha^{fl/fl}$ mice (8-10 weeks, $n=6$ per group) at day 4 post NP-LPS immunization (i.p). NP-specific antibody secreting cells (ASCs) were measured by ELISpot assay.

Reviewer 1 Fig. 25: Flow cytometry analysis of NP-specific PBs ($B220^{low}CD138^{+}NP^{+}$) in the spleen at day 5 post immunization with NP-Ficoll (i.p) ($n=5$ per group).

4. Could you better explain the lower panel of Fig 1D, this is very confusing. Also, does IgM ELISpot assay give similar results in $p38\alpha^{-/-}$ and WT?

Response: We are sorry for the confusion and have re-drawn Fig 1D in the revised manuscript (Reviewer 1 Fig. 26, which is Fig. 1d in the revised manuscript).

As the reviewer suggested, NP-specific ASCs in $p38\alpha^{fl/fl}CD19^{cre}$ and $p38\alpha^{fl/fl}$ mice after NP-OVA/Alum immunization were analyzed. Our results showed that the number of NP-specific IgG1 ASCs was decreased in NP-OVA/Alum-immunized $p38\alpha^{fl/fl}CD19^{cre}$ mice, but the number of NP-specific IgM ASCs was similar in $p38\alpha^{fl/fl}CD19^{cre}$ and $p38\alpha^{fl/fl}$ mice (Reviewer 1 Fig. 27, which is Fig. 1d in the revised manuscript).

Reviewer 1 Fig. 26: NP-specific IgG1 antibody secreting cells (ASCs) in the spleen of $p38\alpha^{fl/fl}CD19^{cre}$ and $p38\alpha^{fl/fl}$ mice (8-10 weeks, n=5 per group) at day 14 post NP-OVA/Alum immunization (i.p) (n=5 per group) were measured by ELISpot assay.

Reviewer 1 Fig. 27: NP-specific IgM antibody secreting cells (ASCs) in the spleen of $p38\alpha^{fl/fl}CD19^{cre}$ and $p38\alpha^{fl/fl}$ mice (8-10 weeks, n=5 per group) at day 14 post NP-OVA/Alum immunization (i.p) (n=5 per group) were measured by ELISpot assay.

5. In the *in vitro* differentiation system, what type of cells are the non-iPC cells? according to the principal component analysis, $p38\alpha^{-/-}$ and WT non-iPCs also exhibited very distinct gene expression profiles, this should be mentioned in the text.

Response: Thanks for this comment and suggestion. The non-iPCs are CD138⁻ B cells in the *in vitro* differentiation system. To make it clear, non-iPCs has now been replaced with CD138⁻ B cells. We have also added a short description of the distinct gene expression profiles of $p38\alpha^{-/-}$ and WT CD138⁻ B cells in the revised manuscript (lines: 159-164 in the revised manuscript).

6. The examination of the involvement of other p38 family members in PC generation, should be mentioned in Supplementary Figure S1. Also, a control showing the lack of expression of the different p38s should be shown, or alternatively refer to the work where these cells have been used before.

Response: Due to the new *in vivo* immunization results were included in the revised manuscript (Supplementary Fig. 2a and 2c in the revised manuscript), the supplementary figures were re-organized. As the reviewer suggested, the examination of the involvement of other p38 family members in iPC generation has been moved to Supplementary Fig. 3 in the revised manuscript to support the specific involvement of p38 α in iPC generation at cell level. The information of p38 family numbers mice used in this study was included in the section of "methods" (lines: 500-501). We examined the protein levels of other p38 family numbers in naïve B cells, and only detected p38 γ , but not p38 β and p38 δ expression by Western blot. Probably p38 β and p38 δ expression in naïve B cells are below the

detection limit since other p38 family members can be detected in C2C12 mouse myoblast cell line (p38 β) or murine pancreas cells (p38 δ) by Western blot. To ensure that the cells lacked the corresponding p38 family members, portions of the CDS sequences of p38 family members in the knockout cells have now been provided (Reviewer 1 Fig. 28, Reviewer 1 Fig. 28b is Supplementary Fig. 3e in the revised manuscript).

Reviewer 1 Fig. 28: (a) Immunoblot analysis of p38 γ and Actin in isolated splenic naïve B cells. **(b)** Portions of the CDS sequences of p38 β , p38 γ and p38 δ of the p38 β ^{-/-}, p38 γ ^{-/-} and p38 δ ^{-/-} mice used in this study.

7. In general, transfection controls are missing. For example, in Figure 2E, 4, 5C... etc controls showing the p38 α , p38 α mutant, BLIMP1 expression are missing.

Response: The expression of p38 α , p38 α mutant, or BLIMP1 have now been included in the corresponding figures in revised manuscript (Reviewer 1 Fig. 29, 30, 31, 32, 33; that are included in Fig. 2e, 4e, 5e, 6c, 6f in the revised manuscript).

Reviewer 1 Fig. 29: Cultured p38 α KO and WT B cells were transduced with retroviruses encoding p38 α and p38 α kinase-dead mutants (K53M or T180A/Y182F) at day 2 of iGCB differentiation. iPCs among retrovirus-infected (CD19⁺GFP⁺CD138⁺) cells were analyzed by flow cytometry, and protein levels of IgG H-chain, p38 α and Actin in retrovirus-transduced cells were analyzed by Western blot 6 days after retroviral transduction.

Reviewer 1 Fig. 30: Immunoblot analysis of BLIMP1, p38α, IgG H-chain, Actin and UPR-related proteins, Bip, IRE1α, SEC61A1 and SEC61B in retrovirally transduced cells (CD19⁺GFP⁺) at day 8 in iPC culture.

Reviewer 1 Fig. 31: Immunoblot analysis of BLIMP1, IgG H-chain, Actin and p38α in TCF3(E47 or E12) overexpressing cells (retrovirally transduced, GFP⁺) at day 8 in iPC culture.

Reviewer 1 Fig. 32: Immunoblot analysis of BLIMP1, IgG H-chain, Actin and p38α in TCF4 overexpressing cells (retrovirally transduced, GFP⁺) at day 8 in iPC culture.

Reviewer 1 Fig. 33: Immunoblot analysis of BLIMP1, IgG H-chain, Actin and p38α in IRF4 overexpressing cells (retrovirally transduced, GFP⁺) at day 8 in iPC culture.

8. Proliferation data presented in Supplementary Fig S3B are not clear, are these cells proliferating? Controls are missing. Do you find the same results using a different method to measure cell proliferation?

Response: We thank the reviewer for this comment. We have now included controls in Fig. S3B of the revised manuscript (Reviewer 1 Fig. 34, which is Supplementary Fig. 4a in the revised manuscript). We have also confirmed that *p38α* deficiency does not affect B cell proliferation in our *in vitro* differentiation system using an EdU staining method (Reviewer 1 Fig. 35).

Reviewer 1 Fig. 34: CFSE-labeled *p38α* KO and WT iGCB cells were transferred onto IL-21 feeder cells. CFSE histogram and MFI of CFSE in iPCs (CD19⁺CD138⁺) and CD138⁻ B cells were analyzed by flow cytometry at indicated time points.

Reviewer 1 Fig. 35: EdU was added into *p38α^{-/-}* and WT B cells cultured on IL-21 feeder cells at 0, 48, 96 hours in iPC differentiation system, and EdU positive cells were analyzed by flow cytometry 2 hours later.

9. On page 8 it would help if the authors explain, in one sentence, the rationale for generating *p38α^{fl/fl}CD19^{cre}Blimp1^{gfp/+}* mouse. On the same page, the authors claim that “*p38α* activation promotes *BLIMP1* expression and plasma cell differentiation”, but what they really show in Fig 3 is a correlation between the expression of *p38α* and a delay in *BLIMP1* expression, so, their data only suggest that *p38α* expression promotes *BLIMP1* expression.

Response: We thank the reviewer for these criticisms and suggestions. *p38α^{fl/fl}Vav^{cre}Blimp1^{gfp/+}* mice would make it easier to monitor PC differentiation, and we have now included this explanation in the revised manuscript (lines: 206-208 in the revised manuscript). We agree with the reviewer that the results in Fig. 3 and have re-worded related sentence (lines: 217-218). As the reviewer suggested in comment 11, we have now moved Fig 5A, 5B, 5C and Supplementary Figure 5 to the section of “*p38α* regulates *BLIMP1* expression and plasma cell differentiation” (Fig. 3e to 3g, Supplementary Fig. 4c to 4e in the revised manuscript). As shown in Fig. 5C of the original manuscript (Fig. 3g in the revised manuscript), ectopic expression of *p38α* kinase-dead mutants failed to restore *Blimp1* transcription in *p38α^{-/-}* B cells (Reviewer 1 Fig. 36), revealing a critical role of *p38α* kinase activity in promoting *Blimp1* transcription and GCB to plasma cell differentiation.

Reviewer 1 Fig. 36: Cultured *p38α* KO B cells were transduced with retroviruses encoding *p38α* or *p38α* kinase-dead mutants (K53M or T180A/Y182F) at day 2 and transferred onto IL-21 feeder cells at day 4. *Blimp1* mRNA levels were measured by quantitative RT-PCR at day 8 of culture.

10. *It would be nice if the authors could explain in the text something about the proteins analyzed in Fig 5E.*

Response: As suggested by the reviewer, a brief description of *p38α* substrates in Fig. 5E has been included in the revised manuscript (lines: 267-271 in the revised manuscript).

11. *I think some results are misplaced, results in Fig 5A, 5B and Supplementary Figure 5 should be moved to the section “p38α regulates BLIMP1 expression and plasma cell differentiation”.*

Response: We agree with the reviewer and have now moved these figures to the section “*p38α* regulates BLIMP1 expression and plasma cell differentiation” in the revised manuscript (Fig. 3e to 3g, Supplementary Fig. 4c to 4e in the revised manuscript).

12. *Is the effect on ER-related transcriptome a consequence of the lower levels of BLIMP1 and decreased differentiation? This should be clarified and discussed in the manuscript.*

Response: Yes, the reviewer is right that the effect of *p38α* on ER-related transcriptome in iPCs was a consequence of lower levels of BLIMP1, as ectopic BLIMP1 expression in cultured *p38α*^{-/-} iPCs restored ER expansion and UPR induction (Reviewer 1 Fig. 37, that are Fig. 4e and 4f in original and revised manuscript), and the ER-related transcriptome was also partially restored by ectopic BLIMP1 expression in cultured *p38α*^{-/-} iPCs (Reviewer 1 Fig. 38, that are Fig. 4c and 4d in original and revised manuscript). We might not describe this clearly and re-wrote this section (lines: 254-261).

Reviewer 1 Fig. 37: (a) Immunoblot analysis of BLIMP1, p38α, IgG H-chain, Actin and UPR-related proteins, Bip, IRE1α, SEC61A1 and SEC61B in retrovirally transduced B cells at day 8 in iPC culture. **(b)** Cultured *p38α* KO and WT B cells were transduced with retroviruses encoding p38α or BLIMP1 at day 2 of *in vitro* differentiation. Representative transmission electron microscopy images of sorted iPCs of indicated groups at day 8.

Reviewer 1 Fig. 38: (a) Heatmap analysis of differentially expressed genes (fold-change > 2)

in iPCs of indicated genotypes. Group 1: decreased in *p38α* KO iPCs, restored by *p38α* or BLIMP1 overexpression. Group 2: decreased in *p38α* KO iPCs, restored by *p38α* but not BLIMP1 overexpression. Group 3: decreased in *p38α* KO iPCs, restored by BLIMP1 but not *p38α* overexpression. Group 4: decreased in *p38α* KO iPCs, restored by neither *p38α* nor BLIMP1 overexpression. Group 5: increased in *p38α* KO iPCs, restored by *p38α* or BLIMP1 overexpression. Group 6: increased in *p38α* KO iPCs, restored by *p38α* but not BLIMP1 overexpression. Group 7: increased in *p38α* KO iPCs, restored by BLIMP1 but not *p38α* overexpression. Group 8: increased in *p38α* KO iPCs, restored neither *p38α* nor BLIMP1 overexpression. **(b)** Gene Ontology enrichment analysis of Group 1 from **(a)**.

13. *Supplementary Figure 5C: Needs quantification to see if protein stability changes or is the same in p38α^{-/-} and WT cells.*

Response: The quantification results have now been included in the revised manuscript as the reviewer suggested (Reviewer 1 Fig. 39, Supplementary Fig. 4e in the revised manuscript).

Reviewer 1 Fig. 39: The densities of the protein bands in Fig. 5C in the previous manuscript (Supplementary Fig. 4e in the revised manuscript) were semi-quantitated.

14. *Could you include the English version of the following webpage? <https://www.cellsignal.cn/learn-and-support/phosphositeplus-ptm-database>*

Response: Sorry about this. The English version of the webpage (<https://www.cellsignal.com/learn-and-support/phosphositeplus-ptm-database>) has now been included in the revised manuscript (lines: 269-270).

15. *Lack of references across the discussion. Also, in the discussion, there are several points: (1) Lines 375-377: could the authors explain which other approaches were used in other studies? (2) Lines 385-386: could the authors explain what they mean in the following sentence? "Thus, the underlying mechanism of aberrant B cells development and GCB cells generation by the treatment of p38 inhibitors needs further investigations" (3) Lines 393-394, in the sentence "Since the germinal center reaction was not well-studied in the study" do the authors mean extensively-studied instead? Please explain (1) and (2) and correct (3).*

Response: We thank the reviewer for these very constructive comments. We added relevant references and have made corrections/deletions in the revised manuscript as

below.

The sentence “Whereas we employed genetic approaches to study the function of p38 α , inhibitors and other approaches were used in previous studies”, and “Thus, the underlying mechanism of aberrant B cells development and GCB cells generation by the treatment of p38 inhibitors needs further investigations” were indeed mixed up. We re-wrote the related sentences (lines: 455-463). The sentence “Since the germinal center reaction was not well-studied in the study” is confusion and we have deleted it in the revised manuscript. As suggested by reviewer 2, we deleted p38 α ,p38 β double KO data in the same paragraphy.

16. Many references are misquoted, these are just two examples: (1) On pages 4 and 5, the authors mentioned that “Previous studies suggested p38 α and p38 β were highly expressed in follicular B cells and p38 α was the most abundant one in PCs” according to references 29 (Khiem et al 2008 PNAS) and 30 (Craxton et al., 1998 J. Immunol); however, in those papers the p38 α and p38 β expression was not described. (2) On methods they claim that p38 α fl/fl, p38 β fl/fl, p38 β -/-, p38 γ -/-, p38 δ -/- mice were previously described in 50 (Wu et al 2013 Cell Res) and 51 (Zheng et al 2011 Nat Cell Biol), this is wrong. Also, the reference Streicher et al., 2014 is missing in the list of references. A general revision of the references is needed in this manuscript.

Response: We apologize for the misquoted references. We have checked all the references and made corrections in the revised manuscript when necessary, including detailed information about the p38 family member knockout mice used in this study.

17. Supplementary Fig S1, explain what is the Pro-pre B cell population.

Response: We are sorry for the confusion. The Pro-pre B cell in our previous manuscript means pro B cells and pre B cells during B cell development (PMID:17923094), and we have changed it to “Pro&Pre B cell” in the revised manuscript. The detailed gating strategy was shown in Reviewer 1 Fig. 40.

Reviewer 1 Fig. 40: The gating strategy to analyze B cell development used in Supplementary Fig. 1.

18. On page 4 line 68 the words “has been used for” are repeated

Response: We are sorry for the mistake and we have now corrected this in the revised manuscript.

Reviewer #2 (Remarks to the Author):

In their study entitled “A p38 α -BLIMP1 signaling pathway is essential for plasma cell differentiation” Wu et al convincingly demonstrate a key role for p38alpha in plasmacytic demonstration. The results are novel and highly relevant, as p38alpha inhibition has therapeutic value in B cell-driven autoimmunity. The experiments are of very high quality and are extremely comprehensive. The authors use relevant in vivo models and take full advantage of a highly suitable in vitro model of plasmacytic differentiation for detailed mechanistic studies.

Wu and colleagues generated mice lacking p38alpha specifically in B cells or in the entire hematopoietic system. They carefully analyzed B cell-dependent immune responses in these mice and found a fundamental defect in plasma cell differentiation and T cell dependent antibody responses, although type II independent antibody responses remained intact. They then employed the 40LB cell culture system to study plasmacytic differentiation in detail. They demonstrated that p38 is the only family member whose sole deficiency deregulates plasmacytic differentiation and that its kinase activity is required. Wu et al show that p38alpha regulates Blimp1 transcriptionally and performed a CRISPR screen of p38alpha targets to identify the relevant downstream mediators. In a series of experiments involving expression of both splice isoforms and knockout they validate Tcf3 as important downstream mediator of p38alpha signaling in plasmacytic differentiation. Finally, Tcf4 and Irf4 are further validated in their role regulating plasmacytic differentiation and Wu et al present evidence for a Tcf3/Tcf4/IRF4-dependent induction of Blimp1 expression downstream of p38alpha. I have only very limited questions and comments as in my opinion this important study can essentially stand as it is.

Response: We thank the reviewer very much for the positive comments.

Major comments

1) What is the role of p38alpha mediated phosphorylation of Tcf3? p38alpha loss does not affect protein levels of Tcf3, Tcf4 and Irf4 only mildly. But overexpression of these proteins compensates for absence of p38alpha in plasmacytic differentiation. Furthermore, the Tcf3-S140A mutant rescues plasma cell differentiation similarly to wild-type Tcf3. Therefore, the role of p38alpha-mediated phosphorylation remains unclear. These aspects are clearly discussed in the manuscript and I do not think the authors have to resolve these questions in the present manuscript. But they should explore whether massive overexpression occurs in their rescue experiments and whether this could explain a (partial) independence of phosphorylation (of Tcf3 or a putative upstream regulator) in this setting. The degree of overexpression of the factors (p38, Tcf3, Tcf4, Irf4) in the rescue experiments should be shown compared to the endogenous levels.

Response: We thank the reviewer very much for the constructive comments. We agree that whether p38 α directly phosphorylates TCF3 is an important question to be addressed. Reviewer 1 also raised a similar question. In the previous version of this manuscript, we showed that the kinase activity of p38 α was critical for BLIMP1 upregulation during iPC differentiation (Fig. 5C in the original manuscript, Fig. 3g in the revised manuscript). The compensation of p38 α deficiency in plasmacytic differentiation by overexpression of TCF3, TCF4, or IRF4 most likely resulted from the basal activities of these transcription factors since reporter gene assays showed transcription factor amount-dependent increase in reporter expression (Reviewer 2 Fig. 1). As suggested by the reviewer, we compared overexpressed TCF3, TCF4, or IRF4 in cultured p38 α ^{-/-} B cells with their endogenous counterparts and showed that the protein levels of overexpressed TCF3, TCF4, or IRF4 were higher than endogenous ones, and BLIMP1 expression was partially restored by TCF3, TCF4, or IRF4 over-expression in cultured p38 α ^{-/-} B cells in iPC differentiation system (Reviewer 2 Fig. 2, 3, 4). These results had been included in Fig. 5e, 6c, 6f in the revised manuscript.

Reviewer 2 Fig. 1: p38^{-/-} 293A cells were transfected with indicated plasmids, the luciferase activity and protein levels of TCF3/TCF4/IRF4 in transfected cells were examined 24 hours after transfection.

Reviewer 2 Fig. 2: Immunoblot analysis of BLIMP1, IgG H-chain, Actin and p38α in TCF3(E47 or E12) overexpressing cells (retrovirally transduced, GFP⁺) at day 8 in iPC culture.

Reviewer 2 Fig. 3: Immunoblot analysis of BLIMP1, IgG H-chain, Actin and p38α in TCF4 overexpressing cells (retrovirally transduced, GFP⁺) at day 8 in iPC culture.

Reviewer 2 Fig. 4: Immunoblot analysis of BLIMP1, IgG H-chain, Actin and p38α in IRF4 overexpressing cells (retrovirally transduced, GFP⁺) at day 8 in iPC culture.

2) GCB cell production seems dramatically enhanced in mice lacking both p38alpha and p38beta in B cells (not slightly as mentioned in the discussion). This comes up only in the discussion and plasma cells are not shown in supplemental figure 8. The authors should either show plasma cells here, as surely they looked at it, or omit this figure as it exceeds the scope of the present manuscript to fully analyze p38alpha/beta double deficiency in B cells.

Response: We thank the reviewer very much for this thoughtful comment. The reviewer is right that “slightly” is not the right word here. B cell specific *p38αp38β* double deficient mice had significantly more GCB cells after immunization (Fig. S8B in the original manuscript, Reviewer 2 Fig. 5). We have also examined antibody secreting cells (ASCs) generation

and antibody responses in NP-OVA/Alum-immunized $p38\alpha^{fl/fl}p38\beta^{fl/fl}CD19^{cre}$ and $p38\alpha^{fl/fl}p38\beta^{fl/fl}$ mice and found that $p38\alpha p38\beta$ double deficient mice exhibited severe defects in ASCs generation and antibody responses (Reviewer 2 Fig. 6). Since the effect of $p38\alpha p38\beta$ double knockout is more complicated than simply enhancing the effect of $p38\alpha$ knockout, we omitted the study of these double knockouts in this manuscript as suggested by the reviewer.

Reviewer 2 Fig. 5: Flow cytometry analysis of GCB cells in the spleen of $p38\alpha^{fl/fl}p38\beta^{fl/fl}CD19^{cre}$ and $p38\alpha^{fl/fl}p38\beta^{fl/fl}$ mice (8-10 weeks, n=6 per group) at day 7.5 after immunization (i.p) with OVA/Alum/LPS. Summary of the percentage and number of GCB cells was shown on the right.

Reviewer 2 Fig. 6: NP-specific IgG1 and IgM antibody secreting cells (ASCs) in the spleen of $p38\alpha^{fl/fl}p38\beta^{fl/fl}CD19^{cre}$ and $p38\alpha^{fl/fl}p38\beta^{fl/fl}$ mice (8-10 weeks, n≥5 per group) at day 14 post immunization with NP-OVA/Alum (i.p) were measured by ELISpot assay.

Minor comments

3) In figure S1 splenic B1 cells should be called B1a, as they are gated on expression of CD5. Also in S8B.

Response: We have corrected the name of these cells in Fig. S1 and S8B in the revised manuscript (Reviewer 2 Fig. 7, 8; that are Supplementary Fig. 1a and 1b in the revised manuscript). Thanks.

Reviewer 2 Fig. 7: (a) Flow cytometry analysis of indicated B cell subsets in the bone marrow and spleen of 5-6 weeks old *p38α^{fl/fl}CD19^{cre}* and *p38α^{fl/fl}* mice ($n \geq 6$ per group). Total B cells: CD19⁺; B2 cells: CD19⁺CD5^{low}CD43^{low}; B1a cells: CD19⁺CD5^{hi}CD43^{hi}; MZ B cells: CD19⁺CD21^{hi}CD23^{low}; Fo B cells: CD19⁺CD21^{low}CD23^{hi}; Mature B cells: B220^{hi}IgM⁺; Immature B cells: B220^{low}IgM⁺; Pro&Pre B cells: B220⁺IgM⁻; Pro B cells: B220⁺IgM⁻CD25⁻CD117⁺; Small pre B cells: B220⁺IgM⁻CD25⁺CD117⁻. **(b)** Flow cytometry of indicated B cell subsets as described in **(a)** in the bone marrow and spleen of 5-6 weeks old *p38α^{fl/fl}Vav^{cre}* and *p38α^{fl/fl}* mice ($n \geq 5$ per group).

Reviewer 2 Fig. 8: Flow cytometry analysis of indicated B cell subsets in the bone marrow and spleen of 5-6 weeks old $p38\alpha^{fl/fl}p38\beta^{fl/fl}CD19^{cre}$ and $p38\alpha^{fl/fl}p38\beta^{fl/fl}$ mice ($n \geq 5$ per group). Total B cells: $CD19^+$; B2 cells: $CD19^+CD5^{low}CD43^{low}$; B1a cells: $CD19^+CD5^{hi}CD43^{hi}$; MZ B cells: $CD19^+CD21^{hi}CD23^{low}$; Fo B cells: $CD19^+CD21^{low}CD23^{hi}$; Mature B cells: $B220^{hi}IgM^+$; Immature B cells: $B220^{low}IgM^+$; Pro&Pre B cells: $B220^+IgM^-$; Pro B cells: $B220^+IgM^-CD25^-CD117^+$; Small pre B cells: $B220^+IgM^-CD25^+CD117^-$.

4) In figure 1D the same data are shown twice with a different scale? Once is sufficient in my opinion.

Response: We are sorry for the confusion and we have now corrected the labels of the Y axis in Fig. 1D in the revised manuscript (Reviewer 2 Fig. 9, which is Fig. 1d in the revised manuscript).

Reviewer 2 Fig. 9: NP-specific IgG1 and IgM antibody secreting cells (ASCs) in the spleen of $p38\alpha^{fl/fl}CD19^{cre}$ and $p38\alpha^{fl/fl}$ mice (8-10 weeks, $n=5$ per group) at day 14 post

immunization with NP-OVA/Alum (i.p) were measured by ELISpot assay.

5) In figure 1E in the secondary immune response the overall IgG1 response (NP29) is lower than the high affinity response (NP7)?

Response: This is a question that we have overlooked. This confusion was caused by the NP7 ELISA KIT we used, as the standard antibody in this KIT is a low affinity NP-specific IgG1, which would lead to overestimation of high affinity NP-specific IgG1 antibody concentrations. We have presented our results as fold-changes in the revised manuscript to avoid confusion (Reviewer 2 Fig. 10, Fig. 1e in the revised manuscript).

Reviewer 2 Fig. 10: $p38\alpha^{fl/fl} CD19^{cre}$ and $p38\alpha^{fl/fl}$ mice were immunized with NP-OVA/Alum (i.p), followed by secondary immunization with NP-OVA (i.p) at day 120 after primary immunization. Serum was collected at indicated time points, and NP-specific antibody titers fold change were determined by ELISA (n=10 per group). Each symbol represents an individual mouse.

6) In figure S2 I would remove the statistical analyses from experiment with n = 2. Also applies to 5J.

S2D: how long were these B cells cultured, please indicate in the figure legend.

Response: We thank the reviewer for the kind suggestion. We have repeated the experiments with $n \geq 5$ and the results were shown in Supplementary Fig. 3c, 3d; Fig. 5f in the revised manuscript (Reviewer 2 Fig. 11, 12). In Fig. S2D, naïve B cells were cultured with BAFF+CD40L+IL-4 for 8 days, cultured with BAFF+CD40L+IL-21 for 6 days, cultured with Anti-IgM+IL-4 for 4 days. This information has been now included in the figure legend of the revised manuscript.

Reviewer 2 Fig. 11: Splenic naïve B cells from $p38\gamma^{-/-}$ (a) or $p38\delta^{-/-}$ (b) and WT mice were cultured in iPC differentiation system. iPCs were analyzed by flow cytometry at day 8. Summary of the percentage of iGCB cells and iPCs.

Reviewer 2 Fig. 12: Flow cytometry analysis of iPCs among cultured $Tcf3$ KO B cells transduced with retroviruses encoding $p38\alpha$. Summary of the percentage of iPCs.

7) Figure 5H: can the luciferase reporter be shown in the supplemental material?

Response: We combined reporter data shown in Fig. 5H, Fig. 6C and Fig. 6G of original manuscript and showed them in Fig. 7a in the revised manuscript.

8) Figure 7B: when were the cells retrovirally transduced to express $p38\alpha$ and its mutants? Same question for 7C.

Response: We are sorry for the incomplete information. Cultured GCB cells were retrovirally transduced to express p38 α and p38 α mutants at day 1 on IL-21 feeder cell in Fig. 7B and 7C in our original manuscript (Reviewer 2 Fig. 13). This information has been included in the figure legend of the revised manuscript.

Reviewer 2 Fig. 13: Schematic outline of *in vitro* plasma cell (iPC) differentiation from splenic GCB cells.

9) Is the full gating strategy to identify B cell subsets always indicated in the figure legends? For example, GCB cells in Figure 1C were only identified by NP, Fas and GL-7?

Response: We are sorry for the incomplete information. Full gating strategy to identify B cell subsets in Fig. 1C was shown as Reviewer 2 Fig. 14 and this information has been included in the related figure legends of the revised manuscript.

Reviewer 2 Fig. 14: The gating strategy to analyze GCB cells in the spleen of immunized mice in Fig. 1C of the original manuscript.

10) Reference 31 indeed shows a reduction in IgG1 levels in p38 deficiency, congruent with the present study. This should be reflected in the discussion.

Response: As suggested, we have now discussed this issue in the discussion section. Reviewer 3 also commented on this issue. As reviewer 3 suggested, we repeated the experiments and replaced the original Fig. S1C as Supplementary Fig. 1c in the revised manuscript (Reviewer 2 Fig. 15).

Reviewer 2 Fig. 15: Serum immunoglobulin (Ig) levels in non-immunized p38 $\alpha^{fl/fl}$ CD19 cre

and $p38\alpha^{fl/fl}$ mice (8-10 weeks, n=8 per group) were determined by ELISA.

Reviewer #3 (Remarks to the Author):

The present manuscript by Wu and Han and their colleagues seeks to understand the role of p38alpha (p38a, MAPK14) in plasma cell (PC) development, as the literature on a possible role of this kinase in this process is unknown. The authors addressed this question by employing a conditional Mapk14 allele encoding p38a and deleting it in B cells (CD19-Cre mice). The main finding that, although germinal center (GC) B cell development is not affected, PC development is markedly impaired, led the authors to elucidate the underlying molecular mechanism in an established in vitro assay that mimics the GC to PC development – both iGC and GC B cells are being cultured and experimentally manipulated. An p38a-BLIMP1 axis with several intermediaries is identified that is critical for the transmission of developmental signals in the GC response and upon LPS stimulation (T-independent type I), but not in the T-independent type II response, demonstrating that not all PC development depends on p38a.

Overall, from Fig. 2 onwards, this manuscript contains high quality data that convincingly uncover a role for p38a in the BLIMP1-dependent development of PCs. This is important information for the field of adaptive immunity in general and PC biology in particular, and the results may be exploited and further developed into potential treatments of PC-associated diseases, including certain autoimmune conditions. However, as detailed below, there are a number of unclear or potentially contentious issues that would require attention.

Response: We thank the reviewer very much for the comments and suggestions, which greatly helped us to improve our studies.

1) On the assumption that in Fig. 1A top (the GC plots), it is gated on B cells, although this is not indicated in the figure legend, the percentage of GC B cells is unusually high, appr. 3 to 4 times higher than expected. Is this because of the unusual OVA/Alumn/LPS conjugate used for the immunization? Why was this conjugate used, whereas in later experiments, only OVA/Alumn was used, which is the normal way (excluding LPS).

Response: We are sorry for the confusion. The reviewer is right that LPS in the OVA/Alumn/LPS precipitate drastically boosts immune responses and leads to the unusually high percentage and numbers of GCB cells and plasma cells. This immunization protocol was originally described in a study by Liu et al (PMID: 27481129) and has been used by other groups. We did not have a special reason to use this protocol but just wanted to use both this enhanced and the standard protocols (OVA/Alumn) in our experiments, and they led to the same conclusion. We are sorry that we did not indicate the gating information, which has been included in the revision. We also mentioned that two protocols were used in the revised manuscript (lines: 105-120).

2) On the assumption that in Fig. 1A bottom (the PC plots), it is gated on splenic mononuclear cells, although this is not indicated in the figure legend, the percentage of

CD138+ plasma cells is unusually high, appr. 5 to 10 times higher than expected. However, CD138+ plasma cells should express 10-higher cell surface amounts of CD138 than in the population shown (one log higher in the FACS plot). Since there is no such population: were the plots gated on B cells, so that PC would be gated out? The bottom line is that as shown, it is not convincing that PC percentages are measured, because there are too many and not at the usual CD138 level. Or does the LPS lead to this phenotype? Unlikely. Some response to these points appears necessary.

Response: We thank the reviewer for these very important questions. We are sorry for the less rigorous gating strategy in the previous manuscript. The gating strategy was: PCs (B220^{low}CD138⁺). We changed our gating strategy and replaced the results in the revised manuscript (Reviewer 3 Fig. 1, which is Fig. 1a in the revised manuscript). As mentioned above, LPS in the OVA/Alum/LPS precipitate drastically boosts immune responses, including GC reaction and PC differentiation (line: 107).

Reviewer 3 Fig. 1: Flow cytometry analysis of GCB cells (CD3-B220⁺Fas⁺GL7⁺) and PCs (B220^{low}CD138⁺) in the spleen of *p38α^{fl/fl}CD19^{cre}* and *p38α^{fl/fl}* mice (8-10 weeks, n>7 per group) at day 7.5 after immunization (i.p) with OVA/Alum/LPS. Summary of the percentage and number of GCB cells and PC was shown.

3) Fig. 1C: What exactly is shown in the plots, what is the gating strategy? This should be indicated in the figure legends and on top of the plots.

Response: The plots in Fig. 1C indicate NP-specific GCB cells and plasmablasts in NP-OVA/Alum-immunized mice. The gating strategy was: NP-specific GCB cells (CD3-B220⁺Fas⁺GL7⁺NP⁺) and NP-specific plasmablasts (B220^{low}CD138⁺NP⁺) and this information has been included in the figure legend of the revised manuscript as suggested by the reviewer.

4) Fig. 1C: CD138+ PCs do not express surface Ig, so what is measured is probably CD138+ plasmablast. So the plots in Fig. 1C bottom are not informative, in contrast to the

ELISPOT shown in Fig. 1D which provide convincing data that PC development is affected.

Response: The reviewer is right and thanks. The description of Fig. 1C has been corrected in the revised manuscript.

5) *Supplementary Fig. 1C: that the basal serum level of IgG1 was decreased in the p38α CD19-conditional mice is in disagreement with the data from the blastocyst complementation study which showed no difference in IgG1 levels in unimmunized mice (ref. 31). However, the control mice have a very large range of serum IgG1 concentrations, and it seems there are actually two subpopulations, with the high concentration ones perhaps representing outliers, letting one wonder if the latter mice were perhaps infected with something? It does not appear that from these results it can be concluded that there is less IgG1 baseline in the conditional p38α mice (see also #8).*

Response: We agree with the reasoning of the reviewer and thus repeated the experiments (Reviewer 3 Fig. 2, which is Supplementary Fig. 1c in the revised manuscript). The baseline of IgG1 was indeed decreased in $p38\alpha^{fl/fl}CD19^{cre}$ mice, though the control mice still exhibited a large range of serum IgG1 concentrations in comparison with KO group. It is hard to make a conclusion on whether there are two subpopulations. The mice were littermates and hosted in the same cages. Decreased numbers of PCs were found in $p38\alpha^{fl/fl}CD19^{cre}$ mice (Reviewer 3 Fig. 3, Supplementary Fig. 1d in the revised manuscript), which may explain the decreased levels of IgG1 in $p38\alpha^{fl/fl}CD19^{cre}$ mice. We carefully re-examined the results in Ref. 31, which also showed a decrease in IgG1 serum level in $p38\alpha$ KO mice. It seems that the big variation in a limited number of mice (n=3) led to the conclusion of no statistical significance in their study (Fig.9 in Ref. 31, PMID:5661878). Nevertheless, we agree with the reviewer that we could err in drawing this conclusion. We have revised the text and considered alternative interpretations (i.e. two subpopulations, contributions of p38α-independent PCs, etc) in the discussion section of the revised manuscript. The original data were replaced with the new results.

Reviewer 3 Fig. 2: Serum immunoglobulin (Ig) levels in non-immunized $p38\alpha^{fl/fl}CD19^{cre}$ and $p38\alpha^{fl/fl}$ mice (8-10 weeks, n=8 per group) were determined by ELISA.

Reviewer 3 Fig. 3: Flow cytometry analysis of GCB cells and PCs in the spleen of non-immunized $p38\alpha^{fl/fl}CD19^{cre}$ and $p38\alpha^{fl/fl}$ mice (8-10 weeks, $n=7$ per group). Summary of the percentage and number of GCB cells and PCs.

6) Fig. 2B: Information on the identity of the differentially expressed genes should be provided in a Supplementary Table. How do we know the analysis was performed correctly? Knowing which genes are differentially expressed may provide answer to this.

Response: As suggested by the reviewer, a supplemental table containing information on differentially expressed genes in $p38\alpha$ WT and KO iGCB cells and iPCs was included in the revised manuscript (Supplementary Table. 1). As expected, $p38\alpha$ was among the most significant differential expression genes in compared $p38\alpha$ WT and KO cells, and the expressions of *Blimp1*, *Xbp1*, *Irf4*, *Bip1* etc, genes important for PCs generation and function, were decreased in $p38\alpha$ KO iPCs compared with $p38\alpha$ WT iPCs (Reviewer 3 Fig. 4). Thus, the analysis results were consistent with our experimental results.

Reviewer 3 Fig. 4: FPKM value of *Blimp1* in and iPCs ($CD138^+$) in the RNA-seq results in Fig. 2b.

7) Discussion line 381: "Gene expression profiles of $p38^{-/-}$ and WT iGCB cells are almost identical, with only 64 differentially expressed genes (Fig. 2B)." Well, 64 genes are quite a lot. Again, here it is important to know the identity of the genes and the fold-differences.

Response: This is because we thought that 64 differentially expressed genes between $p38\alpha$ KO and WT iGCB cells are much less than 1632 differentially expressed genes between $p38\alpha$ KO and WT iPCs. We removed "only" and provided the identity of these 64 genes and the fold-differences between $p38\alpha$ KO and WT iGCB in Supplementary Table. 1 in the revised manuscript.

8) Discussion line 400: Again, the comparison of the serum IgG titers (baseline, without immunization) between ref. 31 and Suppl. Fig. 1C is rendered ambiguous by the apparent two groups in the p38fl/fl (no Cre) control. This could be clarified by repeating the experiment shown in the figure, although overall the data on a role of p38a in the generation of GC derived PCs are so convincing that demonstrating that it does not have effect -- or a small one -- on baseline IgG1, which may well come from different sources of p38a-independent PCs, is in the view of this reviewer not essential.

Response: We thank the reviewer for this comment. As mentioned in our response to comment #5 of this reviewer, we have repeated this experiment and the original figure has been replaced with the new one in the revised manuscript (Reviewer 3 Fig. 2). We have also revised the text accordingly. We agree with the reviewer that the small difference in baseline IgG1 is not an essential question.

9) Introduction line 50, ref. 10: these experiments were done in B-cell lymphoma lines and not mature B cells. This would need to be clarified.

Response: We have changed the description in the revised manuscript as suggested by the reviewer. Thanks.

10) Introduction line 91: Streicher et al. is in the wrong reference format. Perhaps the sentence should be worked over, because to say that "...dysregulated PC differentiation often leads to ... multiple myeloma" is not quite accurate, as myeloma is a tumor of fully differentiated PCs.

Response: We agree and thanks. We have made changes in the revised manuscript as suggested (lines: 86-88).

REVIEWERS' COMMENTS

Reviewer #1 (Remarks to the Author):

the authors have answered all my questions

Reviewer #2 (Remarks to the Author):

In their revision, the authors addressed my (very limited) concerns. I continue to strongly support publication of this comprehensive and important manuscript.

Reviewer #3 (Remarks to the Author):

The authors have thoroughly addressed my concerns and as a result produced a stronger story which is highly relevant for the field of adaptive immunity.

Point-by-point response

REVIEWER COMMENTS

Reviewer #1 (Remarks to the Author):

The authors have answered all my questions.

Response: We thank this reviewer very much for helping us in improving our manuscript.

Reviewer #2 (Remarks to the Author):

In their revision, the authors addressed my (very limited) concerns. I continue to strongly support publication of this comprehensive and important manuscript.

Response: We thank this reviewer very much for helping us in improving our manuscript.

Reviewer #3 (Remarks to the Author):

The authors have thoroughly addressed my concerns and as a result produced a stronger story which is highly relevant for the field of adaptive immunity.

Response: We thank this reviewer very much for helping us in improving our manuscript.